# DNA as a quantum system in evolution

**Nahuel Aquiles Garcia** [ORCID]*

GECORP. Av. Juan Manuel de Rosas 899. S. M. del Monte, Buenos Aires, Argentina

* gecorpinfo@gmail.com

## Abstract

Time may be viewed as an emergent consequence of increasing information entropy. I explore a toy quantum-information model in which DNA is treated as an open quantum system. In this framework, weak, time-dependent perturbations (potentially arising from thermal fluctuations, ionic microfields, metabolic noise, or electromagnetic signals) bias the micro-timing of events during replication and repair. These slight timing shifts can influence the fate of transient electronic and protonic configurations (including short-lived tautomeric states driven by proton-transfer tunnelling), subtly altering mutation probabilities. To test this idea, I map nucleotides in the Mycobacterium tuberculosis genome to constrained qubit states and quantify informational structure using Shannon and von Neumann entropies and coding to non-coding correlation metrics. Simulations of Hamiltonian dynamics under physiologically plausible perturbations show that real genomic segments exhibit distinctive dynamical signatures compared with controls. I also examine a variant in which a weak, slowly varying external signal is introduced as a background "beat" against which DNA dynamics can be compared. Because a Doppler shift in electromagnetic waves encodes the flow of time through the relative motion of source and observer, a cosmic microwave background (CMB) with a tiny frequency drift provides a conceptual clock and a source of informational entropy: it feeds a time-correlated input into the DNA quantum system, allowing the molecule to sample cosmic time and translate it into a biological scale by modulating tunnelling probabilities and thus mutation patterns. This CMB-inspired drive is simply a convenient illustration; the model does not rely on it, and other sources of weakly structured entropy could be tested. Across simulations, sequence-dependent responses to both intrinsic and structured perturbations generate testable predictions: changing the structure or timing of these weak perturbations should produce reproducible shifts in mutation spectra. This framework connects cellular ageing and evolution to the flow of cosmic time and suggests experiments to probe DNA's sensitivity to time-dependent perturbations.

**Data availability statement:** All relevant data are within the manuscript and its Supporting Information files.

**Funding:** The author(s) received no specific funding for this work.

**Competing interests:** The authors have declared that no competing interests exist.

## Introduction

### Evolution as a molecular mechanism

Mutation and selection are the canonical drivers of evolution, but the temporal organization of DNA processes largely determines which errors survive to be inherited [1]. DNA replication leaves behind a transient signature (e.g., hemimethylated sites for strand discrimination) that creates a race between repair initiation and the decay of that signature [1]. Any factor that shifts the relative rates in this race (protein copy numbers, local temperature/ionic micro-environments, chromatin state, or weak electromagnetic inputs) can alter which replication errors are fixed as mutations and therefore reshape the mutation spectrum on which selection acts [2,3].

Quantum effects (e.g., proton tunneling in hydrogen bonds) are one candidate within a broader set of perturbations; they need only operate as one contributor to small timing shifts that become evolutionarily visible over long timescales [4]. This aligns the physics with mainstream evolutionary mechanisms: from variation to bias in timing-sensitive reactions to selection [5,6].

### DNA as a quantum system

Rieper et.al., studied entanglement properties in DNA. By modeling the electron clouds of nucleic acids in DNA as a chain of coupled quantum harmonic oscillators with dipole-dipole interaction between nearest neighbors resulting in a van der Waals type bonding [7], they showed that for realistic parameters nearest neighbor entanglement in DNA is present even at room temperature. Ivan Hubac et. al., studied quantum entanglement and quantum information in biological systems in relation to DNA [8]. They defined qubits formed by Majorana fermions in the hydrogen bonds of DNA (A- T and C-G) and also investigated the entangled states in base pairs. They showed that in hydrogen bonds of the A-T and C-G base pairs of the DNA molecule, the Majorana fermions of the hydrogen electrons are spatially well separated. Recently, Aroche et.al., proposed that DNA operates as a perfect quantum computer [9], leveraging quantum principles to explain the transmission, coding, and decoding of genetic information. It describes DNA's nitrogenous base pairs as forming quantum states, with a Josephson Junction analogy linking these pairs.

Recent work by Mejía-Díaz et al. [10] supports the notion of DNA as a quantum system, proposing that genomic mutations []arise through quantum transitions governed by environmental interactions. Their Hamiltonian-based framework treats DNA as a computational structure capable of processing physical signals.

### DNA as a fractal antenna

Martin Blank and Reba Goodman [11] proposed that DNA is a fractal antenna in electromagnetic fields. They conclude that the DNA's wide frequency range of interaction with electromagnetic fields (EMF) is the functional characteristic of a fractal antenna, and DNA appears to possess the two structural characteristics of fractal antennas, electronic conduction, and self-symmetry. In 2017, Singh P. et al. [12] showed DNA as an electromagnetic fractal cavity resonator. They reported that the 3D-A- DNA

structure behaves as a fractal antenna, which can interact with the electromagnetic fields over a wide range of frequencies. Using the lattice details of human DNA, they modeled the radiation of DNA as a helical antenna. The DNA structure resonates with the electromagnetic waves at 34 GHz, with a positive gain of 1.7 dBi.

## DNA and biophotons

Cells produce photons of biological origin-biophotons (not bioluminescence) related to metabolic activity produced by electronic triplets that participate in the electron transport chain in metabolic processes [13]. The term biophoton refers to ultra-weak photon emissions (UPE) observed in biological systems, especially under metabolic activity. These biophotons are typically in the visible to near-UV range (200–800 nm) and are produced endogenously by organisms as a byproduct of oxidative metabolic reactions, particularly involving reactive oxygen species (ROS) and excited molecular states. These emissions are distinct from quantum-optics-defined biphotons (entangled photon pairs from signal-idler pairs generated via spontaneous parametric down-conversion-SPDC), and while coherence has been reported, it is not in the SPDC sense. Cell-to-cell communication by biophotons has been demonstrated in animal cells. These biophotons are easily detected in the brain (and other tissues of the body) using standard approaches [14]. In 1984, Dr. Popp and colleagues proposed DNA as a source of coherent biophoton emission [15], taking DNA as an exciplex laser system, where a stable state can be reached far from thermal equilibrium at a threshold. In 2021, Na Li. et al. demonstrated that the energy released by phosphoanhydride-bond (PB) hydrolysis of deoxynucleotide triphosphates (dNTPs) during DNA synthesis is in the form of biophotons to drive DNA replication, by modulating their resonance [16].

**Main hypothesis.** DNA is well recognized as a crucial player in the aging process [17–21], which implies an intrinsic relation between DNA and time measures in the cell.

During replication, an error faces (i) a repair-initiation clock (loading/cleavage by mismatch-repair factors) and (ii) a strand-discrimination clock (e.g., decay of hemimethylation or other polarity cues). A mutation is fixed if the second clock "expires" before the first completes; anything that lengthens discrimination or shortens repair raises mutation probability at that site, and the converse lowers it [1]. This timing logic also extends to base-pair tautomerization: short-lived proton-transfer states that last a bit longer during polymerase passage can increase misincorporation odds [4]. External perturbations (quantum effects variations such as probability changes in tunneling occurrence by electromagnetic signals) could alter this timing logic.

Photons can travel to infinity in a hyperbolic geometry without experiencing time [22,23]. Then, as the fractal properties of frequencies are related to the hyperbolic geometry [24,25], which motivates the idea that a fractal biostructure could, in principle, act as a multi-scale antenna. Experiments by Blank and Goodman and others have reported resonant responses of DNA-like systems around 34 GHz with a positive gain, suggesting that hydrated DNA behaves as a fractal antenna in this band [11,12]. Thus, I speculate that DNA could interact with two kinds of signals, the biological biophotonic signals produced by the cells [13–15,26] and another type of photonic signals coming from outside the cells, potentially including cosmic sources. Photon energy at 34 GHz is equivalent to ~$1.4 \times 10^{-5}$ eV (electron volts), which is in the microwave region of the electromagnetic spectrum. At 34 GHz, the photons that DNA might interact with could come from Cosmic Microwave Background (CMB) or solar emissions. The CMB consists of low-energy microwave photons that permeate the universe. While its peak is at 160 GHz, there is still significant intensity in the 30–40 GHz range. The sun emits photons across the electromagnetic spectrum, including the microwave range, especially during solar flares. Thus, DNA structures would have the ability to interact with photonic signals at two different scales, one biological and one universal. If I assume these cosmic signals arrive at Earth from outer space, due to the expansion of the universe, they would be reaching the cells with a wave-phase-shift (Doppler effect). Cosmic signals wave-shift could work as an information entropy input, which DNA would be capable of measuring and transmitting on a biological scale. Establishing a bridge, the DNA would be acting as a cosmic antenna to translate signals from a universal scale to a biological scale, capturing information input about time measure-universe expansion. DNA is able to handle and use information entropy [8–10,27,28].

Therefore, I propose that DNA acts as a quantum system, a quantum computer establishing circuits to operate in a precise frame of reference, which is delimited by the information input in the form of shifted waves coming from outer space. Thus, the Doppler shift resulting from universal expansion provides a naturally occurring, stable, and continuous measure of time that can act as a bridge between cosmic and biological scales. Its predictability, universality, and resistance to local time variations make it an ideal natural clock for living organisms on Earth. This translation mechanism involves the use of cosmic information to alter the probabilities of quantum tunneling in DNA [4], which implies the introduction of mutations in the sequence, a phenomenon that can be understood as a time-measuring process with implications in aging and evolution.

A computer requires by definition "input, processing, and output". Here, input comes from Doppler-shifted cosmic radiation. Processing is performed by DNA's quantum state evolution, governed by both internal Hamiltonians (base-specific binding energies and nearest-neighbor couplings) and external Hamiltonians (cosmic perturbations modeled as time-dependent fields). The output is the altered probability of proton tunneling, leading to mutation. This dynamic quantum response positions DNA as a biological quantum computer that measures and integrates information from universal time flow. In this model, DNA processes informational entropy (e.g., via quantum state variability) without violating the second law of thermodynamics, as energy from cosmic radiation provides the necessary input to sustain local informational order within a globally increasing thermodynamic entropy.

While this study proposes a novel hypothesis, it is crucial to acknowledge the speculative nature of the concept. The framework is rooted in theoretical principles, simulations, and prior evidence, aiming to inspire further empirical validation through experimental research. I propose that **small physical perturbations (here at quantum-scale) can modulate micro-timing windows** in DNA **processes**, biasing mutation fates and regulatory timing in ways that selection can amplify across generations. The speculative cosmology in this prior framing (Doppler-shifted signals) is retained only as an optional source of structured perturbations (informational entropy input); the hypothesis does **not** depend on it. Thus, other sources of perturbations could be tested in future works.

## Alternative external signals beyond the CMB

Here the CMB functions as a spatially uniform template for a weak microwave drive; however, several astrophysical sources deliver structured electromagnetic backgrounds at Earth with Doppler signatures set mainly by relative motion. Solar radio emission spans ≈1 GHz to >100 GHz (quiet Sun and flare gyrosynchrotron bursts) and can transiently dominate the microwave band, providing a natural, time-variable carrier near the 34 GHz motif [29]. The diffuse Galactic radio background and discrete emitters (e.g., pulsars) add additional broadband and quasi-narrowband components whose received frequencies are Doppler-shifted by source kinematics and by Earth's rotation/orbit. A useful "cosmic reference tone" is the neutral-hydrogen 21 cm line at 1420.405751 MHz: it is spectrally sharp, lies in the atmospheric radio window, and is routinely used for Doppler tracking in astronomy [30]. Biophysically, sub-10 GHz candidates are also favored because attenuation rises steeply into the mmWave regime; around 30–40 GHz, gaseous absorption and tissue penetration limits become materially more restrictive (see limitations section) [31].

## Experimental basis for DNA–EM coupling and the "antenna" metaphor

Evidence that DNA can couple to electromagnetic fields is mixed but non-empty. Length-dependent resonant microwave absorption has been reported in aqueous DNA preparations, including claims of direct excitation of internal DNA modes [32,33]. Conversely, other dielectric studies in the 1–10 GHz range reported no enhanced absorption under their conditions, highlighting sensitivity to hydration, ionic strength, conformation, and probe geometry [34]. In the THz band, spectroscopy has resolved conformation-dependent low-frequency modes in DNA oligomers and has distinguished methylation-dependent signatures in genomic DNA [35,36]. Accordingly, "DNA as a fractal antenna" is best treated as a phenomenological shorthand: a hierarchically packed, electrically polarizable polymer (plus hydration shell) can support

multi-scale coupling, with effective coupling set by the dielectric microenvironment rather than by bare bases alone. This framing strengthens the manuscript's central point: a Doppler-variable astrophysical radio background could supply the structured perturbation needed to test timing-modulation mechanisms, without relying exclusively on unattenuated CMB photons at 34 GHz. I cite these studies for context and transparency; however, reported GHz responses are condition-dependent and not uniformly reproduced across setups, so here 'fractal antenna' denotes multiscale EM coupling rather than a definitive in-vivo resonance claim.

To study this hypothesis, I simulated the quantum behavior of the Mycobacterium tuberculosis genome. All modeled quantum phenomena are assumed to occur under physiological temperature conditions, consistent with prior studies showing that DNA-related quantum coherence and tunneling can persist at biological temperatures [4,7].

## Materials and methods

All the analyses were performed over the *Mycobacterium tuberculosis* genome and its annotations (https://www.ncbi.nlm.nih.gov/datasets/genome/GCA_013010385.1/). All the developed codes are available at https://github.com/nahuelaq-uiles/-DNA-as-system-computer-to-measure-time. Please see Supporting information for an extended detailed version of the methods.

A)   Shannon entropy

The complete Mycobacterium tuberculosis genome was segregated into coding and noncoding regions based on the described annotations. Shannon entropy was calculated for each region using:

$$H(X) = -\sum_{\{i=1\}}^{n} p\left(x_i\right) \backslash log_2 p\left(x_i\right)$$

H(X) is the Shannon entropy of the sequence X.
$p(x_i)$ is the probability (relative frequency) of the *i*-th nucleotide in the sequence.
$n$ is the number of unique nucleotides in the sequence (e.g., A, T, C, G).
The logarithm is in base 2 to reflect the entropy in bits.
Steps Applied:

1. Count the frequency of each nucleotide in the sequence.

2. Convert counts to probabilities by dividing by the total sequence length.

3. Sum the entropy contributions for all nucleotides, each weighted by $-p(x_i) \log\_2 p(x_i)$.

B) Von Neumann entropy

The von Neumann entropy is a key measure of quantum informational diversity. I computed the quantum entropy by first constructing the density matrix (ρ) for the qubits of each region. The DNA to qubit mapping strategy is described in the results.

The von Neumann entropy is then computed using the density matrix ρ:

$$\rho = \frac{1}{n}\sum_{i=1}^{n} |\psi_i\rangle\langle\psi_i|$$

$$S(\rho) = -\sum_{i} \lambda_i \log_2 \lambda_i$$

n is the number of nucleotides in the sequence.

$\langle \psi_i \rangle$ is the quantum state of the i-th nucleotide, determined by the DNA-to-qubit mapping. Von Neumann Entropy ($S(\rho)$): Once the density matrix is constructed, the entropy is calculated. Lambda_i ($\lambda_i$) are the eigenvalues of the density matrix. Eigenvalues represent the probabilities of different quantum states, and their logarithmic contribution measures uncertainty.

DNA-to-quantum-state encoding: Let $s = s_1 \dots s_N$ be a DNA sequence of length N. I map each nucleotide $s_i \in \{A,C,G,T\}$ to a 2-component (qubit) state vector $|\varphi(s_i)\rangle \in C^2$ by:

- $|\varphi(A)\rangle = (1, 0)^T$ ($|0\rangle$)

- $|\varphi(T)\rangle = (0, 1)^T$ ($|1\rangle$)

- $|\varphi(C)\rangle = (1, 1)^T / \sqrt{2}$ (($|0\rangle + |1\rangle)/\sqrt{2}$)

- $|\varphi(G)\rangle = |vac\rangle$ (erasure/vacuum state). In the implementation I denote this by the zero vector $(0, 0)^T$; it contributes a zero projector and the final density matrix is renormalized to unit trace.

Region density matrix: For each non-vacuum site i, define the pure-state projector $\rho_i = |\varphi(s_i)\rangle\langle\varphi(s_i)|$. Let $N\_eff$ be the number of non-vacuum sites. The region density matrix is defined as the average:

$$\rho = \frac{1}{N_{eff}} \sum_{i:s_i \neq G} \rho_i$$

By construction $\rho$ is positive semidefinite; I enforce $Tr(\rho)=1$ (equivalently, $\rho \leftarrow \rho/Tr(\rho)$).

Von Neumann entropy is then computed as:

$$S(\rho) = -Tr\left(\rho \log_2 \rho\right) = -\sum_k \lambda_k \log_2\left(\lambda_k\right)$$

where $\{\lambda_k\}$ are the eigenvalues of $\rho$ and the sum includes only $\lambda_k > 0$ (numerically, $\lambda_k < 10^{-12}$ are ignored).

Illustrative example. For s = "ATC":

$$\rho_A = \begin{bmatrix} 1 & 0 \\ 0 & 0 \end{bmatrix}, \ \rho_T = \begin{bmatrix} 0 & 0 \\ 0 & 1 \end{bmatrix}, \ \rho_C = \frac{1}{2}\begin{bmatrix} 1 & 1 \\ 1 & 1 \end{bmatrix}$$

Thus $\rho = (1/3)(\rho_A + \rho_T + \rho_C) = [[0.5, 1/6],[1/6, 0.5]]$. The eigenvalues are $\{2/3, 1/3\}$, giving $S(\rho) = -(2/3)\log_2(2/3) - (1/3)\log_2(1/3) \approx 0.9183$ bits.

Site-basis complex-amplitude encoding (used for Schrödinger evolution in Sections D–F). For a region of length N, I initialize a state vector $\psi(0) \in C^N$ in the site basis by an equivalent scalar embedding $\alpha(b)$:

$$\alpha(A) = 1, \ \alpha(T) = i, \ \alpha(C) = \frac{1+i}{\sqrt{2}}, \ \alpha(G) = 0$$

I set unnormalized amplitudes $\psi_i(0) = \alpha(s_i)$ and normalize $\psi(0) \leftarrow \psi(0) / \|\psi(0)\|_2$. This scalar encoding is consistent with the qubit mapping above via the identifications $|0\rangle \to 1$, $|1\rangle \to i$, $(|0\rangle + |1\rangle)/\sqrt{2} \to (1+i)/\sqrt{2}$, and $|vac\rangle \to 0$.

C) Entanglement Analysis Between Coding and Non-Coding Regions

1. Data Source and Preparation

First 5 coding and non-coding regions were selected for the entanglement analysis to ensure computational efficiency. Sequences were extracted using 0-based indexing to align with Python's slicing behavior.

2.   Qubit Mapping Strategy

Each DNA nucleotide was mapped to a qubit state based on previous evidence [7–9]:
To translate DNA sequences into quantum states:
Mapping Strategy:

- A (Adenine): [1, 0] ($|0\rangle$)

- T (Thymine): [0, 1] ($|1\rangle$)

- C (Cytosine): [1, 11] (superposition state $|0\rangle + |1\rangle$)

- G (Guanine): [0, 0] (collapsed state)

3.  Construction of the Joint Density Matrix

For each coding and non-coding pair, a joint density matrix was constructed using the tensor product of the individual qubit states. The joint density matrix captures the entangled states between the two regions.

$$\rho_{joint} = \frac{1}{n_1 n_2} \sum_{i=1}^{n_1} \sum_{j=1}^{n_2} |\psi_i \otimes \phi_j\rangle\langle\psi_i \otimes \phi_j|$$

Where: n1 and n2 are the number of qubits in the coding and non-coding regions, respectively. $|\psi_i\rangle$ and $|\phi_j\rangle$ are the qubit states of the i-th coding nucleotide and j-th non-coding nucleotide. $\otimes$ represents the tensor product.

4.   Partial Trace and Reduced Density Matrix

To analyze the entanglement between coding and non-coding regions, I computed the partial trace of the joint density matrix. The partial trace gives the reduced density matrix for one region by summing over the states of the other region.
$\rho_A = Tr_B(\rho_{joint})$
Where: $\rho_A$ is the reduced density matrix for the coding region. $Tr_B$ represents the partial trace over the non-coding region's states.

5.   Entanglement Entropy Calculation

The von Neumann entropy was used to quantify entanglement. It measures the amount of quantum information shared between the two regions. See Supplementary information for a detailed description.

D) Quantum state evolution over time without external perturbations.

Parameters estimation
To calculate the period of the 34 GHz wave: The frequency (f) is 34 GHz, which means: T = 1 / f = 1 / (34 × 10^9 Hz) ≈ 2.941 × 10–11 seconds (29.41 picoseconds). To set the time step size (dt) to accurately capture the oscillations, the time step (dt) should be an order of magnitude smaller than the period. Thus, dt = T / 100 ≈ 2.941 × 10–13 seconds.
Modeling approach
Quantum System Initialization: the DNA sequence was separated into two segments: coding and non-coding regions, representing distinct structural and functional areas. Each nucleotide was assigned a specific quantum amplitude (previously described mapping strategy)

**Hamiltonian Construction:** to simulate DNA's internal dynamics, a Hamiltonian matrix was constructed for each region. The diagonal elements represent the binding energies of each nucleotide, randomly sampled from a range of 0.01 to 0.05 eV. Off-diagonal elements denote coupling strengths, set to 0.025 eV, allowing for nearest-neighbor quantum interactions.

**Quantum Evolution:** using the constructed Hamiltonians, time evolution was simulated over 100,000 steps with a time interval (dt) of $1 \times 10^{-13}$ seconds. The evolution was performed by calculating the exponential of the Hamiltonian, $U = \exp(-iH \cdot dt/\hbar)$, applied iteratively to the initial state vector. States were normalized at each step to maintain probability conservation.

Data Analysis: after simulation, amplitudes and phases were extracted from the quantum states for statistical comparison. Statistical tests (T-test and Mann-Whitney U) compared the coding and non-coding regions' amplitude and phase distributions.

E) Quantum state evolution over time with external perturbations

System Setup and Sequence Segmentation: a DNA sequence was divided into non-coding and coding regions. Each nucleotide was mapped as previously described. This mapping initializes the quantum state vector, enabling the modeling of quantum dynamics specific to each region.

Hamiltonian Construction: a Hamiltonian matrix was constructed for each region, where diagonal elements represent binding energies, assigned between 0.01 and 0.05 eV[7–9]. Off-diagonal elements were set to 0.025 eV, representing nearest-neighbor coupling. These Hamiltonians provide a base for simulating DNA's intrinsic quantum interactions.

Quantum Evolution with External Perturbation: quantum evolution was modeled over 100,000 steps with a time interval (dt) of $1 \times 10^{-13}$ seconds. A time-dependent perturbation, with an amplitude (λ) of $1 \times 10^{-41}$ eV and frequency of 34 GHz, was added to the Hamiltonian at each time step, simulating the effects of external cosmic radiation. The Hamiltonian at each time step was adjusted using this perturbation term, and the system's state was updated iteratively with a first-order approximation.

Data Extraction and Statistical Analysis: Amplitudes and phases were computed from the quantum states across time, and time-averaged values were calculated. Statistical comparisons between the coding and non-coding regions' amplitudes and phases were conducted using both T-tests and Mann-Whitney U tests.

Introducing wave shift effect: a perturbation was applied to simulate external cosmic interactions, with an initial frequency of 34 GHz and a linear frequency shift rate (k) of $-1 \times 10^{18}$ Hz/s to model wave shifts over the simulation period. Separate Hamiltonians were constructed for the non-coding and coding regions. Diagonal elements, sampled from 0.01 to 0.05 eV, represent nucleotide binding energies, while off-diagonal elements of 0.025 eV account for nearest-neighbor interactions. The Hamiltonians were then combined into a single matrix with an additional coupling term (0.01 eV) between the two regions, simulating inter-region interaction. For each time step, the perturbation was modified according to the Doppler shift, adjusting the frequency by $k \cdot t$, where t is the time step in seconds. This generated a time-dependent Hamiltonian, enabling continuous interaction between the non-coding and coding regions. Phases were detrended to isolate oscillatory components before applying Fourier Transforms to both non-coding and coding regions' phase data, highlighting frequencies within the expected Doppler shift range. This spectral analysis identifies amplitude differences across the wave-shifted frequency range. Additionally, power spectra within the Doppler-shifted frequency range were calculated for both regions, and a T-test was applied to the power values to identify significant differences.

Parameters Selection:

Set k = -1e18 Hz/s to produce a significant frequency shift over the simulation time.

Over the simulation time (num_steps * dt), this results in a frequency shift from 34 GHz to approximately 33 GHz.

**Notes on Realistic Parameters:** the actual Doppler shift due to the universe's expansion over human timescales is negligible. For simulation purposes, I introduce a larger frequency shift to observe the effects on the DNA system.

**Incorporated Antenna Gain:** Incorporated antenna gain: the reported gain of 1.7 dBi was converted to a linear scale using gain_linear = 10^(gain_dBi/10). The perturbation amplitude was then rescaled as λ_perturbation *= gain_linear to mimic an enhanced effective coupling between the external field and the DNA-like system. In the present work this is used as a generic fractal-antenna-inspired correction and should not be interpreted as a quantitatively calibrated coupling to cosmological radiation.

To avoid implementation-style descriptions in the main text, the coupled coding/non-coding evolution is defined below purely in mathematical terms; construction details, numerical integration choices, and validation are reported in the Supplementary Methods.

Hilbert space and basis:

Let $\mathcal{H} = \mathbb{C}^N$ be the state space in a site (nucleotide) basis $\{|i\rangle\}_{i=1}^N$, with $N = N_{nc} + N_c$. I partition the sites into a non-coding block $\mathcal{H}_{nc} \cong \mathbb{C}^{N_{nc}}$ (sites $i = 1, \ldots, N_{nc}$) and a coding block $\mathcal{H}_c \cong \mathbb{C}^{N_c}$ (sites $i = N_{nc} + 1, \ldots, N$), so that

$$\mathcal{H} = \mathcal{H}_{nc} \oplus \mathcal{H}_c.$$

A general (pure) state is

$$|\psi(t)\rangle = \sum_{i=1}^N \psi_i(t)|i\rangle, \qquad \psi(t) = (\psi_1(t), \ldots, \psi_N(t))^\mathsf{T} \in \mathbb{C}^N.$$

Total Hamiltonian:

I model coherent evolution with a (generally) time-dependent Hamiltonian $H_{tot}(t) \in \mathbb{C}^{N \times N}$, decomposed as

$$H_{tot}(t) = \underbrace{(H_{nc} \oplus H_c)}_{\text{uncoupled blocks}} + \underbrace{H_{couple}}_{\text{boundary coupling}} + \underbrace{H_{pert}(t)}_{\text{external drive}},$$

where $H_{nc} \in \mathbb{C}^{N_{nc} \times N_{nc}}$ and $H_c \in \mathbb{C}^{N_c \times N_c}$. In the site basis, the direct sum $H_{nc} \oplus H_c$ is the block-diagonal matrix

$$H_{nc} \oplus H_c = \begin{pmatrix} H_{nc} & 0 \\ 0 & H_c \end{pmatrix}.$$

Boundary-only inter-region coupling:

To impose minimal interaction between the two subspaces, I couple only the boundary sites $i = N_{nc}$ and $j = N_{nc} + 1$. Let $g \in \mathbb{C}$ denote the coupling strength (in energy units). I define

$$H_{couple} = g|N_{nc}\rangle\langle N_{nc} + 1| + g^*|N_{nc} + 1\rangle\langle N_{nc}|,$$

so that, equivalently, the only nonzero off-diagonal matrix elements introduced by $H_{couple}$ are

$$(H_{couple})_{N_{nc}, N_{nc}+1} = g, \qquad (H_{couple})_{N_{nc}+1, N_{nc}} = g^*,$$

and all other entries are zero.

Time-dependent perturbation restricted to the non-coding block.

I apply a diagonal driving term only on $\mathcal{H}_{nc}$. Let $\lambda \in \mathbb{R}$ be its amplitude. Define a (slowly varying) instantaneous frequency

$$f(t) = f_0 + kt,$$

with $f_0 \geq 0$ and chirp rate $k$. Then

$$H_{\text{pert}}(t) = \lambda \cos(2\pi f(t)t) \sum_{i=1}^{N_{\text{nc}}} |i\rangle \langle i|.$$

In matrix form, $H_{\text{pert}}(t)$ is diagonal with entries $\left(H_{\text{pert}}(t)\right)_{ii} = \lambda \cos\left(2\pi f(t)t\right)$ for $1 \leq i \leq N_{\text{nc}}$ and $\left(H_{\text{pert}}(t)\right)_{ii} = 0$ for $i > N_{\text{nc}}$.

Initialization from nucleotide identity:

Given a sequence $\{b_i\}_{i=1}^{N}$ with $b_i \in \{A, T, C, G\}$, define a complex-valued encoding $\alpha : \{A, T, C, G\} \to \mathbb{C}$. In this work I use

$$\alpha(A) = 1, \quad \alpha(T) = i, \quad \alpha(C) = \frac{1 + i}{\sqrt{2}}, \quad \alpha(G) = 0.$$

I set the unnormalized initial amplitudes by

$$\widetilde{\psi}_i(0) = \alpha\left(b_i\right), \qquad i = 1, \ldots, N,$$

and normalize to obtain

$$\psi(0) = \frac{\widetilde{\psi}(0)}{\|\widetilde{\psi}(0)\|_2}.$$

Consistently with the block structure, I write

$$\psi(0) = \begin{pmatrix} \psi_{\text{nc}}(0) \\ \psi_{\text{c}}(0) \end{pmatrix} \in \mathbb{C}^{N_{\text{nc}}} \oplus \mathbb{C}^{N_{\text{c}}}.$$

Continuous-time dynamics:

The evolution is given by the time-dependent Schrödinger equation

$$i\hbar \frac{d}{dt}|\psi(t)\rangle = H_{\text{tot}}(t)|\psi(t)\rangle, \qquad |\psi(0)\rangle \text{ as above.}$$

Formally, the solution can be written using the time-ordered propagator

$$|\psi(t)\rangle = \text{Texp}\left(-\frac{i}{\hbar} \int_0^t H_{\text{tot}}(s)\, ds\right) |\psi(0)\rangle.$$

Numerical integration:

For numerical experiments, I discretize the **time-dependent Schrödinger equation**

$$i\hbar \frac{d\psi(t)}{dt} = H_{\text{tot}}(t)\, \psi(t)$$

on a grid $t_n = n\Delta t$. A first-order explicit step reads

$$\psi\left(t_{n+1}\right) \approx \left[I - \frac{i\Delta t}{\hbar} H_{\text{tot}}\left(t_n\right)\right] \psi\left(t_n\right),$$

while the exact one-step unitary update for a frozen Hamiltonian is

$$\psi\left(t_{n+1}\right) = \exp\left(-\frac{i}{\hbar}H_{\text{tot}}\left(t_n\right)\Delta t\right)\psi\left(t_n\right).$$

Implementation details (matrix construction, time stepping, renormalization/diagnostics, and validation checks) are provided in the Supplementary Methods.

Interpretation of the coupling term:

The inter-region boundary coupling implemented in $H_{\text{tot}}$ (i.e., the off-diagonal interface terms $H_{N_{nc}, N_{nc}+1} = H_{N_{nc}+1, N_{nc}} = g_{nc,c}$) provides the minimal mechanism by which amplitude can transfer between the non-coding block $H_{nc}$ and the coding block $H_c$.

**Continuous Evolution Function:** combined Hamiltonian (H_total). The key to introducing entangled properties in the code is the construction of a combined Hamiltonian H_total, which represents the entire DNA system comprising both non-coding and coding regions. This includes the Hamiltonians of the non-coding (H_nc) and coding (H_c) regions are placed along the diagonal of H_total. A coupling between the non-coding and coding regions is introduced at the boundary between them. This coupling term represents the interaction that can lead to entanglement.

H_total[len_nc − 1, len_nc] = H_total[len_nc, len_nc − 1] = coupling_nc_c

Here, coupling_nc_c is a parameter representing the strength of the interaction between the last nucleotide of the non-coding region and the first nucleotide of the coding region.

The quantum state of the entire system is represented by a state vector state_total, which combines the states of the non-coding and coding regions:

state_total = np.zeros(total_length, dtype=complex)
state_total[:len_nc] = initialize_state(non_coding_sequence)
state_total[len_nc:] = initialize_state(coding_sequence)

Each nucleotide in the DNA sequence is mapped to a complex amplitude, representing its quantum state.

Site-basis complex-amplitude encoding (used for Schrödinger evolution in Sections D–F). For a region of length N, I initialize a state vector $\psi(0) \in C^N$ in the site basis by an equivalent scalar embedding $\alpha(b)$:
$\alpha(A) = 1$, $\alpha(T) = i$, $\alpha(C) = (1 + i)/\sqrt{2}$, $\alpha(G) = 0$.

I set unnormalized amplitudes $\psi_i(0) = \alpha(s_i)$ and normalize $\psi(0) \leftarrow \psi(0) / \|\psi(0)\|_2$. This scalar encoding is consistent with the qubit mapping above via the identifications $|0\rangle \to 1$, $|1\rangle \to i$, $(|0\rangle + |1\rangle)/\sqrt{2} \to (1+i)/\sqrt{2}$, and $|vac\rangle \to 0$.

**Time Evolution with Coupling:** the time evolution of the system is governed by the Schrödinger equation, discretized for numerical simulation:

state_total = state_total − (1j * dt / hbar) * np.dot(H_t, state_total)
state_total /= np.linalg.norm(state_total)

**Perturbation on Non-Coding Region:** A time-dependent perturbation, simulating the cosmic signals with a Doppler shift, is applied only to the non-coding region:

H_t[:len_nc,:len_nc] += np.diag(lambda_perturbation * np.cos(2 * np.pi * frequency_t * time) * np.ones(len_nc))

**Interaction through Coupling:** The coupling term in H_total allows the perturbation effects in the non-coding region to influence the coding region during the time evolution. This is because the Hamiltonian includes off-diagonal elements that connect the states of the non-coding and coding regions.

F)  Computational model simulating proton dynamics in DNA: tunneling

I model proton transfer within a base pair by a one-dimensional effective coordinate $x$ along a hydrogen-bond direction, with a symmetric quartic double-well potential. This is a simplified model intended to capture generic tunneling behavior rather than fully asymmetric, environment-dependent potentials of specific base pairs.

## Hamiltonian and dynamics

For each base pair I consider a wavefunction $\psi(x, t) \in L^2(\mathbb{R})$ evolving under the time-dependent Hamiltonian

$$\hat{H}(t) = \hat{T} + \hat{V} + \hat{V}_{\text{ext}}(t), \qquad \hat{T} = -\frac{\hbar^2}{2m_p}\frac{d^2}{dx^2}, \qquad \left(\hat{V}\psi\right)(x) = V(x)\psi(x),$$

where $m_p$ is the proton mass. The evolution is governed by

$$i\hbar\frac{\partial}{\partial t}\psi(x, t) = \hat{H}(t)\psi(x, t), \qquad \int_{-\infty}^{\infty}|\psi(x, t)|^2\,dx = 1.$$

Quartic double-well potential.
  I define

$$V(x) = V_0\left[\left(\frac{x}{a}\right)^4 - 2\left(\frac{x}{a}\right)^2 + 1\right],$$

which has minima at $x = \pm a$ and a barrier of height $V_0$ at $x = 0$.
  External driving and noise.
  I add a weak perturbation consisting of a deterministic drive plus Gaussian noise,

$$V_{\text{ext}}(x, t) = (\lambda + \xi(t))\cos(2\pi f(t)t)\frac{x}{a},$$

with amplitude $\lambda$ and zero-mean noise $\xi(t)$. I use a slowly varying instantaneous frequency

$$f(t) = f_0 + kt.$$

  Initial condition
  I initialize a normalized Gaussian wavepacket localized in the left well:
$$\psi(x, 0) = \left(\frac{1}{\pi\sigma^2}\right)^{1/4}\exp\left(-\frac{(x-x_0)^2}{2\sigma^2}\right), \qquad x_0 = -a.$$
  Tunneling probability
I quantify tunneling by the probability of finding the proton in the right well,

$$P_R(t) = \int_0^{\infty}|\psi(x, t)|^2\,dx.$$

Split-operator propagation (details in Supplementary Methods).
  For numerical propagation I use Strang splitting over a time step $\Delta t$,

$$\psi(x, t + \Delta t) \approx \exp\left(-\frac{i}{\hbar}\hat{T}\frac{\Delta t}{2}\right)\exp\left(-\frac{i}{\hbar}(\hat{V} + \hat{V}_{\text{ext}}(t))\Delta t\right)\exp\left(-\frac{i}{\hbar}\hat{T}\frac{\Delta t}{2}\right)\psi(x, t).$$

All discretization and implementation choices are reported in the Supplementary Methods.

1.  Overview of the Computational Model

The simulation models proton transfer between nucleotides in a base pair using a one-dimensional effective double-well potential representing the hydrogen bond. This potential is deliberately chosen as a symmetric quartic toy model, designed

to capture generic tunnelling behaviour rather than to reproduce the detailed, asymmetric and environment-dependent proton-transfer potentials of specific G–C or A–T pairs. Within this simplified framework, the model incorporates:

• Construction of double-well potentials for each base pair.

• Initialization of the proton's wave function in one of the wells.

• Time evolution of the proton's wavefunction using the split-operator method to solve the time-dependent Schrödinger equation.

• Inclusion of an external perturbation representing incoming cosmic signals with a frequency shift due to the Doppler effect from the universe's expansion.

• Introduction of Gaussian noise to simulate stochastic resonance, enhancing the system's response to weak signals.

• Calculation of the tunneling probability over time for both coding and non-coding DNA regions.

• Statistical analysis to compare tunneling probabilities between coding and non-coding regions and control sequences.

2.  Simulation parameters and variables

### 2.1. Physical constants

Reduced Planck's constant ($\hbar$): $1.0545718 \times 10^{-34}$ J•s
Proton mass ($m_p$): $1.6726219 \times 10^{-27}$ kg
Elementary charge (e): $1.6021766 \times 10^{-19}$ C

### 2.2. Spatial Grid

Number of spatial points (N): 512
Spatial domain (x-grid): from $-5 \times 10^{-10}$ m to $5 \times 10^{-10}$ m
Spatial step size (dx): calculated from x-grid, approximately $1.96 \times 10^{-12}$ m

### 2.3. Time Parameters

Time step (dt): $1 \times 10^{-15}$ s
Number of time steps (num_steps): 100,000
Total simulation time: dt×num_steps = $1 \times 10^{-10}$ s (0.1 ns)

### 2.4. External Perturbation Parameters

Antenna gain in decibels isotropic (gain_dBi): 1.7 dBi
Linear gain (gain_linear): calculated as 10^(gain_dBi/10)
Electric field amplitude ($E_0$): $1 \times 10^{-6}$ V/m
Characteristic length scale (a): $1 \times 10^{-10}$ m
Perturbation amplitude ($\lambda_{perturbation}$): $eE_0 a \times$ gain_linear

### 2.5. Frequency Parameters

Base frequency ($f_0$): 34 GHz, chosen as a representative GHz-scale drive motivated by reports aof DNA sensitivity around this frequency; in the model it plays the role of a generic external resonance, not a literal in vivo CMB frequency.
Frequency shift ($\Delta f$): 0.0008 Hz, which generates a very slow chirp over the simulated time window. This value is of the same order as that expected for cosmological Doppler shifts at 34 GHz over a few hours, but here it is used purely as a mathematical template for a slowly varying, structured drive. 2.6. Noise Parameters
Noise amplitude (noise_amplitude): 0.05 (dimensionless scaling factor)

3.  DNA Sequences and Regions

A real DNA sequence from the M. tuberculosis genome of 200 base pairs (bp) was selected, consisting of Non-coding region (first 100 bp) and coding region (last 100 bp). The complementary sequence was generated using standard base-pairing rules (A-T, C-G). A control sequence of 200 bp was generated, entirely composed of non-coding DNA. In the simulation code, the control sequence was artificially divided into 'coding' and 'non-coding' regions to serve as a baseline comparison.

4.  Construction of the Double-Ill Potential

The base pair is formed by a nucleotide from the sequence and its complement. Parameters are assigned based on the base pair type. A-T or T-A pairs. Potential barrier height ($V_0$): 0.065 eV. Characteristic length (a): $1 \times 10^{-10}$ m. G-C or C-G pairs. Potential barrier height ($V_0$): 0.108 eV. Characteristic length (a): $1 \times 10^{-10}$ m

4.3.  Potential Function

The effective double-well potential V(x) is defined as: $V(x) = V_0[(x/a)^4 - 2(x/a)^2 + 1]$
   This symmetric quartic form creates two minima (wells), separated by 2a, representing two stable proton positions along a generic hydrogen bond.
   In real DNA, proton-transfer potentials for A–T and G–C base pairs are known to be asymmetric and to depend sensitively on the local environment (hydration shell, neighbouring bases, proteins, ions). Quantitatively accurate potential energy surfaces for these processes are typically obtained using density functional theory (DFT) or quantum mechanics/molecular mechanics (QM/MM) approaches that explicitly include the surrounding atoms. Performing such calculations for each base-pair context along an extended genomic sequence, and then propagating the resulting open quantum dynamics under a time-dependent GHz drive over biologically relevant timescales, is currently computationally prohibitive. For this reason, in the present work I adopt the symmetric quartic potential as a tractable effective model. The parameter $V_0$ is chosen to be of the same order of magnitude as barrier heights reported in more detailed ab initio studies, but the resulting tunnelling rates, wavefunctions and level splittings should be interpreted only as qualitative outputs of this toy model. The goal is to investigate how a weak, structured perturbation (with a Doppler-like frequency evolution) can modulate tunnelling probabilities and how these modulations differ between real and shuffled DNA sequences, rather than to make quantitative predictions for in vivo proton transfer."

5.  Initialization of the Proton's Wave Function

The proton's initial wave function ψ(x,t=0) is set as a Gaussian wavepacket localized in one of the wells (e.g., the 'left' well):
   Standard deviation (σ): $5 \times 10^{-11}$ m
   Initial position ($x_0$): -a (left well)
The wave function is normalized to ensure total probability equals one.

6.  Time Evolution of the Proton's Wave Function

The time evolution of the proton's wave function is governed by the time-dependent Schrödinger equation. The Hamiltonian (H) consists of kinetic and potential energy terms $H = T + V(x)$.
   Kinetic Energy Operator (T): Represented using finite differences in the spatial domain.
   Potential Energy Operator (V(x)): Includes the double-well potential and external perturbations.
   The external perturbation $V_{ext}(x,t)$ simulates the effect of incoming cosmic signals with a frequency shift:
   $V_{ext}(x,t) = (\lambda_{perturβation} + \xi(t)) \cos(2\pi(f_0 + \Delta f)t) (x/a)$
   $\lambda_{perturβation}$: Perturbation amplitude due to cosmic signals.

ξ(t): Gaussian noise term to simulate stochastic resonance.

$f_0 + \Delta f$: Frequency of the incoming signal with Doppler shift.

Stochastic Resonance as Gaussian noise with zero mean and specified amplitude is added to the external perturbation to simulate environmental fluctuations enhancing the detection of weak signals. The split-operator method is employed to numerically solve the time-dependent Schrödinger equation. Momentum space transformation applies the kinetic energy operator in momentum space using the Fourier transform. Potential energy application applies the potential energy operator in position space. Alternate between kinetic and potential operators for each half-time step to achieve second-order accuracy. The wave function is periodically normalized to prevent numerical errors.

7.  Calculation of Tunneling Probability

At each time step, the probability of the proton being in the 'right' well is calculated:

This probability is recorded over the entire simulation time for each base pair.

8.  Simulation Execution

Simulations are conducted separately for non-coding and coding regions in both real and control sequences. For each region, the proton tunneling probability is computed for every base pair, resulting in a set of tunneling probability curves over time. The average tunneling probability across base pairs and runs is calculated for each region.

Note: Please check Supplementarylementary Methods for a detailed description (including the mathmathical framework) used across the manuscript.

## Results

To test these ideas, different calculations, mathematical models, and computational simulations were applied to the Mycobacterium tuberculosis genome. The coding regions make up 4,034,041 base pairs (bp), while the non-coding regions account for 394,851 (8.92%).

### Shannon entropy

Shannon entropy offers a robust way to quantify the informational diversity within DNA sequences. By measuring the entropy of coding and non-coding regions, I explored how DNA handles informational complexity and coherence. The calculated Shannon entropy for the tuberculosis genome is on average 1.92 in coding regions and 1.81 in non-coding region entropy (T-test 18.01 | P $5.03 \times 10-69$) (Fig 1A and 1B). Lower entropy in non-coding regions suggests less informational diversity in these sequences. Non-coding regions might act as regions where coherent quantum states are maintained for longer periods, analogous to memory registers or low-noise quantum channels. Higher entropy in coding regions aligns with the notion of complex biological processing, as coding regions need to handle diverse genetic instructions for protein synthesis, introducing variability that increases entropy. The significant entropy difference supports the thesis that DNA operates as a dual-mode quantum system where non-coding regions maintain coherence.

Assuming that DNA is able to function as a quantum computer [9], mapping DNA sequences to qubits enables the translation of nucleotide patterns into quantum states, allowing for quantum entropy calculations. This approach is crucial for modeling DNA as a quantum system, helping to explore how coding and non-coding regions interact through quantum principles. To map the DNA sequence to qubits, each base (A, T, C, G) was represented as a qubit [9]. This approach models the qubit states based on binary representations for simplicity.

Mapping Strategy:

A (Adenine): [1, 0] ($|0\rangle$)

T (Thymine): [0, 1] ($|1\rangle$)

C (Cytosine): [1, 1] (superposition state $|0\rangle + |1\rangle$)

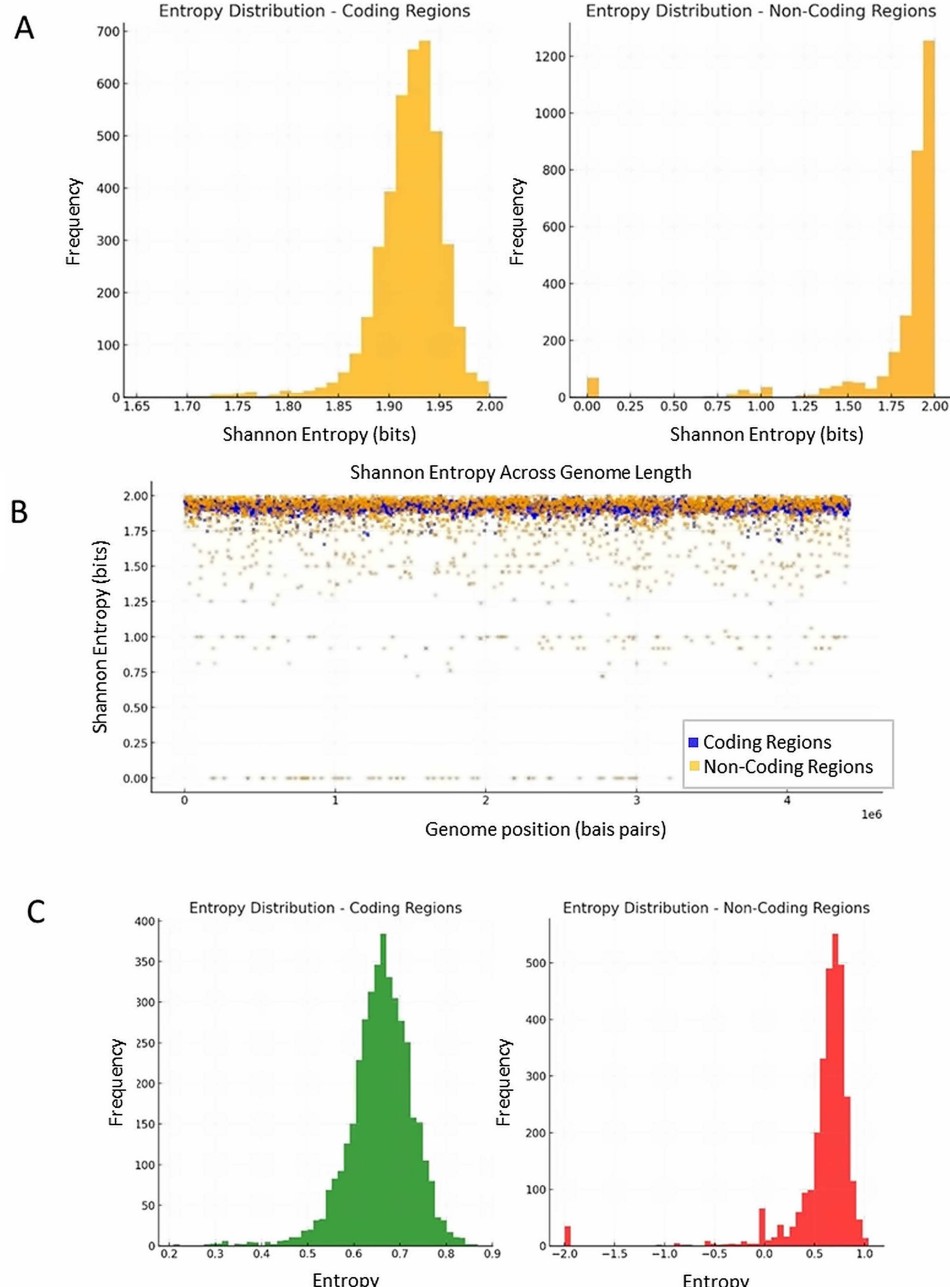

**Fig 1. Comparative entropy and information diversity between coding and non-coding regions of DNA. (A)** Shannon entropy shows a significant difference between coding and non-coding regions. Coding regions exhibit a higher average Shannon entropy (1.92 bits) compared to non-coding regions (1.81 bits). Statistical significance was determined (T-test: p = 5.03 × 10-69), underscoring robust differences in informational complexity. **(B)** Distributions of Shannon entropy values for coding and non-coding regions. Coding regions exhibit a narrower distribution with lower variability, suggesting consistent information density. Non-coding regions show broader variability, potentially reflecting their dual roles as stabilizing buffers and adaptive sensors for external signals. **(C)** Von Neumann entropy values. The plot illustrates the distinct distributions of quantum entropy for the two types of regions. Coding regions have higher and more consistent entropy values (mean: 0.6576, lower standard deviation), reflecting their role in managing stable and structured quantum information for biological processes. Non-coding regions display lower mean entropy (0.5925) but greater variability (T-test: p = 7.73 × 10-22).

G (Guanine): [0, 0] (collapsed state)

This nucleotide-to-qubit mapping reflects discrete, biologically plausible quantum states based on prior models of DNA quantum behavior [1–3]. While not spanning the full Bloch sphere, it allows simulation of constrained quantum dynamics, consistent with the hypothesis that DNA evolved to exploit quantum coherence in a biologically adapted subspace (see limitation section).

The mapping above defines a deterministic encoding map M: Sigma -> H from the four-letter DNA alphabet Sigma = {A, C, G, T} into a low-dimensional quantum state space. It is applied independently at each position i, so any DNA sequence s = s1...sN of arbitrary length is mapped by applying M(si) to every nucleotide and then building (i) a site-amplitude state vector used in the Hamiltonian simulations (Methods) or (ii) a product/tensor representation used for the density-matrix calculations (Supplementary Methods). Thus, the approach is general with respect to DNA sequence; the "four cases" simply reflect the four symbols of the nucleotide alphabet.

A discrete subset of states is used. Although a qubit admits a continuum of pure states on the Bloch sphere, DNA in vivo is an open quantum system. Environmental coupling (hydration shell, ions, thermal noise) is expected to rapidly decohere most superpositions, effectively selecting a restricted set of robust pointer-like states. In this toy model I therefore start from a minimal set of basis/superposition states that can be interpreted as coarse-grained electronic/protonic configurations, while the subsequent Hamiltonian evolution still produces continuous complex amplitudes and phases. This choice is also conservative: it avoids over-parameterizing the encoding in the absence of experimentally validated nucleotide-specific Bloch-sphere angles.

Construction of the guanine "collapsed" state: The vector [0, 0] is not a normalized qubit and is used here as shorthand for an erasure/vacuum state |vac> orthogonal to {|0>, |1>}. Computationally, guanine sites are initialized with zero amplitude and the full state is normalized, so guanine contributes no projector term |psi><psi| at initialization. Conceptually, this represents a strongly damped/background degree of freedom within the chosen coarse-graining (i.e., a state that is rapidly relaxed/dephased and therefore does not carry coherent amplitude in this encoding).

Generalized mapping (explicit). For completeness, the encoding can be generalized to assign any nucleotide (or local k-mer context) to an arbitrary qubit state on the Bloch sphere. For each base b in {A, C, G, T}, define

$$\left| \psi(b) \right\rangle \; = \; \cos\left( \frac{\theta_b}{2} \right) \; \left| 0 \right\rangle \; + \; e^{i\, \varphi_b} \sin\left( \frac{\theta_b}{2} \right) \; \left| 1 \right\rangle$$

where (theta$_b$, phi$_b$) are fixed parameters or can be derived from normalized biochemical descriptors (e.g., stacking energies, hydrogen-bond pattern, dipole moment, oxidation potential) and optionally extended to local sequence context. This parameterized option is described in the Supplementary Methods as an alternative to the discrete encoding used in the main simulations

For each region of the genome, sequences were extracted and converted into qubit representations. These qubits were then used for subsequent Von Neumann entropy calculations. Coding regions showed a mean entropy of 0.6576, while the mean value for non-coding regions was 0.5925 (Fig 1C) (P-value for t-test: $7.73 \times 10-22$). Coding regions show more consistent entropy values with lower standard deviation, suggesting more uniform informational content. Non-coding regions exhibit greater variability in entropy, possibly reflecting the presence of stabilizing, buffer-like sequences alongside regions of randomness or low density. Coding regions with lower variability and more consistent entropy values may act as stable quantum states that support the precise computation of biological rhythms. Non-coding regions variability could reflect their higher sensitivity to external perturbations, and may act as adaptive quantum sensors, capturing and processing fluctuations from external sources.

## Entanglement between coding and non-coding regions

Analyzing entanglement between DNA regions allows to understand the interplay between coding segments, acting as stable quantum circuits, and non-coding segments, functioning as adaptive sensors. Given that the main hypothesis

posits that DNA synchronizes biological rhythms with cosmic timelines, entanglement entropy serves as a direct indicator of how quantum correlations within DNA enable this adaptive synchronization. If non-coding regions are the main structures to capture cosmic signals and transfer that quantum information to coding regions, it suggests that the entire DNA system operates as a coherent quantum mechanism, adjusting internal biological processes to align with external, cosmic changes. Measuring entanglement properties provides quantifiable evidence of shared quantum information and coherent interactions.

To measure the degree of quantum correlation between both genome regions, I treated the DNA sequences as quantum systems, mapping each nucleotide to a qubit state (same Mapping DNA sequences to qubits strategy as for quantum entropy calculation) and then calculating entanglement entropy to quantify shared quantum information. I explored how the quantum states (qubits) of each region relate to one another. The technical approach included mutual density matrices and entanglement entropy. The calculated average entanglement entropy between the first 5 coding and non-coding regions was 0.75. This value provides insights into the degree of quantum correlation between the two regions. This indicates that the first 5 coding and non-coding regions have moderate to high entanglement, suggesting that these regions are not independent but quantum-correlated. This implies that information from non-coding regions influences the states of coding regions, and vice versa. Thus, the quantum connection between regions supports the idea that DNA functions as an adaptive quantum system, where both region types interact to maintain coherence.

## Quantum state evolution over time: implicit perturbations

This section delves into how DNA's quantum states evolve in response to internal, implicit perturbations, without external influences. Assaying quantum state evolution over time could reveal how DNA's coding and non-coding regions dynamically respond to quantum interactions. To test it, I applied a quantum modeling strategy based on the Schrödinger equation to simulate the dynamics of DNA sequences. The DNA sequence under study was divided into non-coding and coding regions, allowing to explore how each region responds to quantum interactions. The quantum states of each DNA base were initialized according to a specific encoding scheme (same mapping DNA sequences to qubits strategy as before). I defined a Hamiltonian to model interactions among the bases and used iterative time evolution to observe how the quantum states changed across different time steps. The model included:

1. Hamiltonian-Driven Evolution: The Hamiltonian (H) in the simulation describes how the quantum states of the system evolve over time, reflecting the energy interactions in the system and governing the time evolution of the quantum states via the Schrödinger equation. For neighboring base interactions, the Hamiltonian incorporates interactions between adjacent nucleotides, which implicitly generates oscillatory and fluctuating behavior in the quantum states.

2. Superposition and Quantum States: The non-coding and coding regions were initialized with different quantum states. Cytosine (C), which represents a superposition state, inherently leads to greater uncertainty and variability in the system.

This approach captures the oscillatory behavior of the quantum states. Firstly, I estimated realistic parameters for the model (see methods section), including time step size (dt) and Hamiltonian (H). The initialized state function for the described mapping strategy assigns each nucleotide a specific complex amplitude as previously demonstrated [9]. This approach lets the evolution dynamics reflect any intrinsic properties tied to the nucleotide types. Thus, this initialization provides a unique "signature" for each nucleotide.

To quantify the differences between the coding and non-coding regions, I calculated the amplitudes and phase shifts of the quantum states at each time step. The analysis revealed amplitude differences (T-test, p-value = 2.76e-56) between the coding and non-coding regions (Fig 2A). This could mean that these regions process quantum information differently. While coding regions might be more "stable" in their amplitudes, potentially reflecting a regulatory or information storage

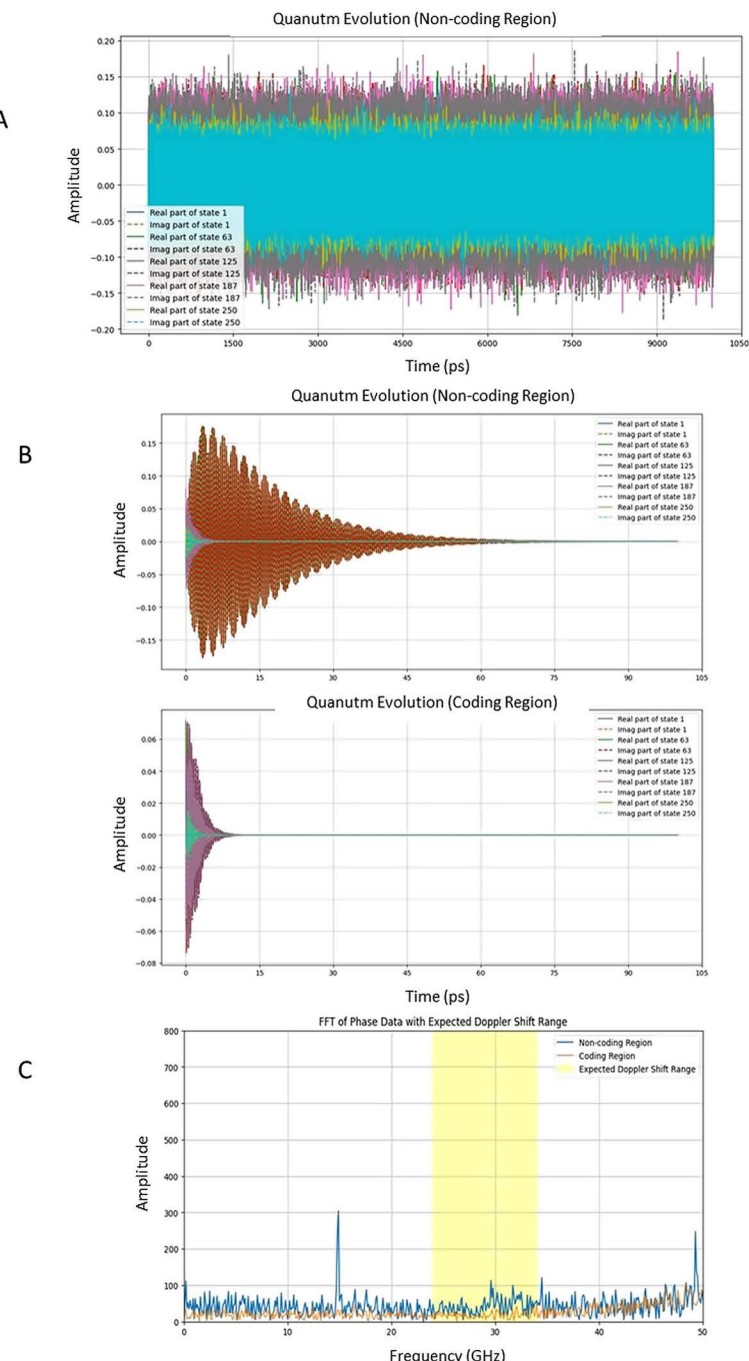

**Fig 2. Quantum state evolution of DNA under varying conditions of cosmic perturbations. (A)** Quantum amplitude distributions without external perturbations. Coding regions (not shown) exhibit more stable and uniform amplitude values, reflecting their role as stable quantum circuits. Non-coding regions display broader variability in amplitudes (T-test, p = 2.76 × 10^-56). **(B)** Evolution of quantum states under external perturbations (34 GHz cosmic radiation), simulated over the first 1,000 steps. **(C)** Power spectrum analysis of phase oscillations under external perturbations with a simulated Doppler-shifted 34 GHz signal. The Fast Fourier Transform (FFT) was applied to the phase data to compute the spectral power, highlighting how DNA regions respond to specific frequency ranges. The average power ratio (real data: 2.77, control: 1.32) showed that different DNA regions have distinct behavior.

role, non-coding regions may display higher variability, possibly acting as "quantum sensors" in adapting to changes. Interestingly, the lack of phase difference might indicate that both regions share a synchronized quantum coherence (without external perturbations). This could mean that, although coding and non-coding regions differ in their amplitude behavior, they maintain a unified quantum phase. This synchronization is crucial for coherent quantum systems, where phase alignment enables communication and information transfer across different parts of the system.

Different dt and time steps were simulated with similar results. Control simulations for false coding and noncoding regions were run (for example, a whole noncoding region was introduced in the program and segregated as coding and noncoding – not shown). Corroborating the hypothesis, results showed a much lower or any difference between regions. By the aleatory properties of the Hamiltonian, several simulations were run for each condition.

### Quantum state evolution over time: external perturbations

This section models the quantum state evolution of DNA under external perturbations by introducing 34 GHz cosmic radiation into the Hamiltonian as time-varying fields. Using the Schrödinger equation, the simulations track how coding and non-coding regions respond over thousands of time steps. Introducing external perturbations via a time-dependent Hamiltonian is a standard method for simulating the effect of external fields. To introduce the 34 GHz perturbation into the Hamiltonian, a practical approach is to represent the external radiation as a time-dependent perturbation. Firstly, I determine the Perturbation Amplitude ($\lambda$), which is a scaling factor that modulates the strength of the 34 GHz field on the Hamiltonian. The energy of a photon at 34 GHz is $\approx 1.406 \times 10-4$ eV. Thus, the chosen perturbation amplitude, $\lambda = 1 \times 10-4$ eV, is justified based on physical considerations. However, considering the interaction strength and the fact that DNA may have resonant absorption at this frequency, using $\lambda$ in the range $10-5$ eV to $10-31$ neV is reasonable for exploratory simulations. In these simulations, I ran 100.000 steps ($dt = 1 \times 10-13$) which resulted in a total simulation time of 10 nanoseconds ([Fig 2B]). This is appropriate for observing quantum dynamics at molecular scales.

After running more than 50 simulations, the amplitude distributions between coding and non-coding regions were not significantly different. In contrast to the simulation without external perturbations (where amplitudes differed between regions), this lack of difference suggests that the external radiation (34 GHz) equalizes amplitude dynamics across coding and non-coding regions. Amplitude differences may signify that, without perturbations, coding, and non-coding regions have intrinsic, structure-dependent dynamical behaviors. The external perturbation, however, seems to "flatten" or "homogenize" these behaviors, potentially aligning both regions to resonate or absorb external frequencies in a uniform way, a process that may support DNA's hypothesized role in cosmic signal measurement.

In the phase analysis, results showed statistically significant phase differences between coding and non-coding regions (T-test, $p = 9.86 \times 10-43$), suggesting that coding and non-coding regions respond to the external perturbation with distinct phase dynamics. Nevertheless, when control experiments using artifact control coding and noncoding regions were applied to the model (ej. analyze an entire noncoding region as coding and noncoding, or vise versa), the simulations showed similar results as real sequences, suggesting that the observed differences could be associated with the nature of the modeling approach (especially due to the random variability introduced by the Hamiltonian). However, as the main hypothesis is related to features of DNA to capture the wave shift (Doppler effect-universe expansion) in incoming entropy information from cosmic signals, the next step was to model and simulate these phenomena.

By incorporating a time-dependent frequency shift, I modeled the Doppler effect of cosmic signals due to universe expansion. The expansion of the universe causes a very slight increase in the wavelengths of cosmic signals I observe. Over short periods like 3 hours, this effect is approximately a 0.0008 Hz frequency at 34 GHz. For initial testing, over the simulation time, I tested frequency shift from 34 GHz to approximately 33 GHz. For simulation purposes, and due to computational limitations, I introduce a larger unrealistic frequency shift to observe the effects on the DNA system. The model and simulation setup followed the previous results which suggested that the non-coding regions could interact with cosmic signals. The coupling (entanglement) between non-coding and coding regions allows for ongoing transfer of information.

In the code, this interaction is modeled by coupling the Hamiltonians of the non-coding and coding regions. The coupling allows quantum information (amplitudes and phases) to be transferred between the regions, simulating entanglement and information flow. This represents the bridging of cosmic and biological time scales through continuous interaction. The DNA system's response to the time-varying frequency simulates the measurement of cosmic time expansion. Thus, analyzing the system's dynamics could reveal patterns corresponding to the Doppler shift. I ran 10 simulations using real DNA sequences, including coding and noncoding regions, and as a control, I ran 10 simulations using a whole noncoding segment but divided the code into coding and noncoding (S1 and S2 Tables). For each region, the amplitude and phase were recorded over time, and statistical tests were applied. Moreover, the power within the Doppler-shifted frequency range was calculated separately for non-coding and coding regions. Fourier transforms were applied to the phase data to analyze the spectral power within the Doppler-shifted frequency range (centered around 34 GHz). The power within the target frequency range provides a measure of how strongly each region responds to the Doppler-shifted signal. In this context, "power" refers to the spectral power within a specific frequency range for the phase data of quantum states in the DNA simulation. Thus, power represents the intensity or "strength" of oscillations (in this case, the phase variations) at certain frequencies. Higher power means stronger oscillations or fluctuations within the chosen frequency band.

The average power ratio in the control runs (1.32) was significantly lower than the average power ratio in the real runs (2.77) (Fig 2C). This indicates that, in the control condition, the non-coding regions did not show a strong distinction in power compared to coding regions. The lower power ratio and smaller standard deviation suggest less variability in the control condition, supporting the hypothesis that the real runs (with a genuine coding/non-coding distinction) exhibit unique behavior. Both the T-test and Mann-Whitney U tests for amplitude and phase differences still show significant p-values in the control runs. However, in the context of the control sequence, this could be due to random fluctuations rather than meaningful differences, especially given the artificial designation of coding/non-coding regions.

The simulations support the hypothesis that non-coding DNA regions act as receivers for cosmic information, which they may transfer to coding regions through quantum entanglement. Control simulations confirmed that these effects were not due to random variations. The coherence results show that non-coding and coding regions maintain a synchronized oscillation in specific frequency bands, suggesting that the phase relationship between the oscillations in both regions is stable and consistent over time. This stable phase relationship implies that any oscillatory signal captured by the non-coding regions is mirrored in the coding regions.

The $\Delta f \approx 0.0008$ Hz shift over 3 hours corresponds to the **cosmological redshift** drift expected from $f \propto 1/\alpha(t)$, i.e., $\Delta f/f \approx H0\Delta t$, rather than the much larger **kinematic Doppler** terms ($\Delta f/f \approx v/c$) associated with Earth's rotation/orbit that would dominate for discrete emitters such as the Sun or narrow astronomical lines.

## Mutations induction by quantum tunneling

Theoretically, quantum tunneling in DNA facilitates proton transfer along hydrogen bonds in nucleotide pairs, creating temporary tautomeric states [4]. These states, if stabilized, can induce mutations, introducing variability in the genetic code. By methodological limitations (see methods), in this work I do not attempt to compute quantitative in vivo tunnelling rates for specific base pairs using fully realistic density functional theory (DFT) or quantum mechanics/molecular mechanics (QM/MM) potentials. Instead, I use a simplified, symmetric double-well model of a hydrogen bond to explore, at a conceptual level, how a weak GHz-range perturbation with a Doppler-like frequency evolution, applied on top of thermal noise at 37 °C, could modulate tunnelling probabilities over time. Introducing somatic mutations in DNA is a well-recognized aging mechanism (time measure) [17–21]. Therefore, I studied how the perturbations in the DNA quantum system induced by incoming shifted cosmic signals could influence the introductions of mutations in the sequence (via quantum tunneling), bridging time measures from universal to biological scale.

I studied if cosmic signals could accumulate enough energy in DNA to influence quantum tunneling and, thereby, affect mutation rates. The activation energy for proton tunneling in DNA (e.g., in G-C base pairs) is estimated at around

0.1–0.5 eV[4], enough to enable tautomeric shifts and possible mutations. I estimate how much energy a single nucleotide absorbs per second from cosmic radiation. Based on cosmic microwave background (CMB) radiation at 34 GHz, each photon has an energy of about $1.4 \times 10^{-5}$ eV, which converts to approximately $2.2 \times 10^{-24}$ joules (J) per photon. Since DNA could act as a fractal antenna, each nucleotide might absorb only a very tiny fraction of this energy. Each nucleotide could absorb $\sim 10^{-32}$ J/s from CMB radiation (calculated using a CMB energy density of $10^{-14}$ J/m³ and nucleotide cross-sectional area of $10^{-18}$ m²). Studies on DNA's fractal antenna behavior suggest that DNA's structure has resonant properties at 34 GHz, meaning it can interact more effectively with frequencies in this range, capturing small amounts of energy over time. Thus, each nucleotide's antenna-like behavior could contribute a small amount of energy individually. However, in a sequence with millions of nucleotides, these small contributions add up cumulatively. Assuming that a DNA nucleotide chain can act cooperatively, where each nucleotide acting as an individual antenna can additively add up to amplify the signal, for example, a DNA sequence with 3 million nucleotides, the total energy absorbed per second is $3 \times 10^{-26}$ J/s. For proton tunneling in DNA, an activation energy threshold of approximately 0.1 eV is necessary. Thus, 0.1 eV corresponds to $1.6 \times 10^{-20}$ J. With these parameters and low interference assumptions, the time needed for the DNA sequence to accumulate this energy (t=Activation Energy/Energy Absorbed per Second) is around 6 days.. These are extremely optimistic assumptions (ignoring atmospheric and tissue attenuation, and neglecting competing thermal and anthropogenic fields, see limitation section). This back-of-the-envelope calculation therefore reinforces, rather than contradicts, the conclusion that CMB photons cannot constitute a dominant direct energy source for proton tunnelling in vivo. In the present work, I interpret cosmological radiation only as a conceptual lower-bound, structured component of the ambient GHz field, whose temporal pattern could in principle bias transition probabilities in a system that is already driven primarily by local thermal and biochemical energy. This is a conceptual model in which I used CMB radiation as a source of external perturbations, laying the foundations for future works to explore other types of signals when analyzing DNA as a quantum system.

Results showed that quantum entanglement properties allow it to maintain coherence DNA when capturing shifted cosmic signals. DNA could utilize this phase-shifted cosmic information to measure time continuously. This proposed time-measurement mechanism bypasses direct energy accumulation by focusing on phase coherence and frequency shifts to synchronize DNA's biological clock with cosmic time.

To investigate the potential role of DNA as a fractal antenna capturing cosmic signals and its impact on proton tunneling (and consequently mutational events), I developed a computational model simulating proton dynamics in DNA hydrogen bonds under the influence of external realistic shifted perturbations (incoming cosmic signals at 34 GHz shifted 0.0008 Hz in 3 hours based on real Doppler effect by universe expansion) and stochastic resonance (SR). The purpose of introducing stochastic resonance was to simulate a realistic biological environment where weak signals could be enhanced by noise. Since the cosmic signal perturbation is extremely weak, without SR, it would have little or no effect on the tunneling dynamics. By introducing noise, SR allows the system to reach a resonance condition where the combined effect of the noise and weak external signal can lead to observable fluctuations in tunneling probabilities. By introducing biological realism, the SR effect makes the system more sensitive to small changes, as noise effectively "pushes" the proton's wave function between the wells, enhancing tunneling probabilities.

I performed multiple simulations comparing real DNA sequences to control sequences that artificially designated all nucleotides as non-coding, regardless of their actual biological roles (Fig 3). The results were statistically analyzed to determine if meaningful differences exist between the tunneling probabilities in real and control sequences. The results of the tunneling probability distributions for coding and non-coding regions in real DNA sequences showed statistically significant differences (S3 Table). In contrast, the control sequences, which lack a genuine biological distinction between coding and non-coding regions, exhibited no consistent or significant differences in tunneling probabilities.

The paired t-test yielded a t-statistic of 3.805 with a p-value of approximately 0.004, indicating that the differences in tunneling probabilities between real and control sequences are significant. The control simulations displayed lower

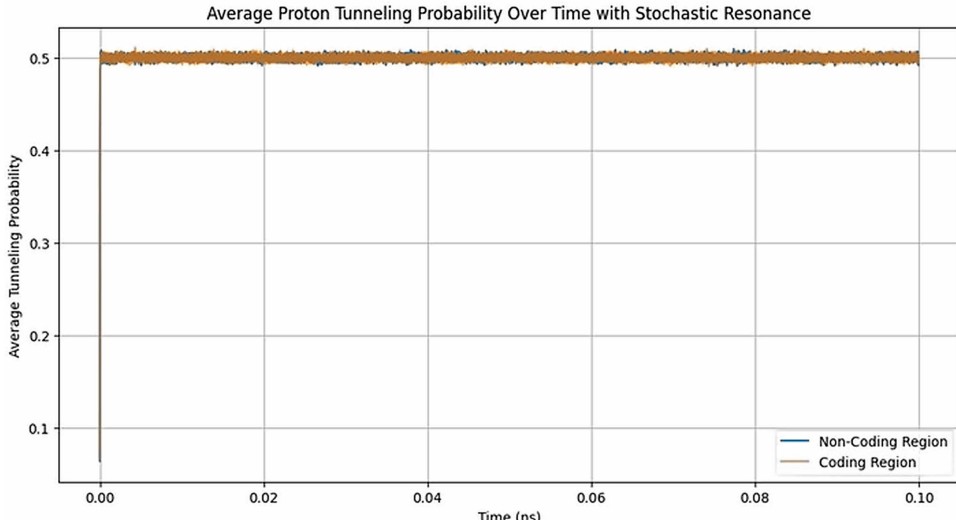

**Fig 3. Proton tunneling dynamics in DNA under external 34 GHz perturbations with a Doppler-shifted wave (0.0008 Hz).** The plot illustrates the evolution of proton tunneling probabilities within coding and non-coding regions.

t-values that lacked consistent significance, suggesting that any differences in these sequences could be attributed to random chance rather than meaningful biological structure. The significant differences observed in real sequences are attributable to the actual biological distinctions between coding and non-coding regions. The differences in tunneling probabilities in real DNA imply that coding and non-coding regions behave differently at the quantum level. Variations in proton tunneling probabilities could influence mutation rates via quantum tunneling mechanisms, potentially impacting aging (time measure) and other biological processes. The fact that significant differences are only observed in real sequences suggests that the structural and informational properties of DNA are crucial for this quantum interaction, potentially enabling DNA to act as a receiver of cosmic shifted information. Finally, I performed the same analysis for the CRY1 gene of humans (S4 Table). The results showed the same patterns as those of Mycobacterium tuberculosis genes, suggesting that the behaviors found in DNA when treated as a quantum system are a conserved mechanism across species.

In summary, these results show that, within this conceptual toy model, a very weak, structured external drive with a Doppler-like temporal evolution can influence tunnelling probabilities within DNA's hydrogen bonds. Any real cosmological contribution to the ambient field would be vastly subdominant to thermal and anthropogenic sources, so the model should be read as a proof-of-principle that sequence structure can support differential responses to such drives. In this sense, quantum-induced mutations in the model can be associated with ageing (time measures) or adaptive processes, but I regard the link between cosmological information entropy and biological evolution as speculative and not yet established experimentally.

## Discussion

This work is based on a speculative hypothesis. With the current state of knowledge, the results cannot be interpreted as direct evidence of cosmological influence on DNA. Accordingly, the present work should be understood as a conceptual quantum model and numerical sensitivity analysis, grounded in real genomic sequences and physiologically motivated noise, rather than as a quantitatively calibrated biophysical model of cosmological radiation acting on DNA.

By analyzing the genome of Mycobacterium tuberculosis, this work provides evidence supporting the main hypothesis. While the hypothesis that DNA operates as a quantum system remains speculative [9], recent experimental findings on

quantum coherence in biological systems lend partial support [7–9,11,12]. This work integrates established principles of quantum biology with theoretical models to explore the potential quantum computational behaviors of DNA.

Shannon entropy showed different values between coding and noncoding regions. This reflects biological systems' need to shift between quantum possibilities and classical realities. This could imply translating cosmic information into concrete biological actions (e.g., cellular functions or adaptation).

The observed differences in the quantum entropy distributions suggest coding and non-coding regions complement each other, where coding regions could function like the "clockwork" of the DNA-based quantum computer, providing stable, ordered computation, while non-coding regions act as receptors for fluctuating cosmic frequencies, analogous to wave shift detection. Thus, the dynamic interaction between these two types of regions could allow DNA to adjust biological time based on quantum perturbations, such as cosmic wave shift signals. The consistent entropy in the coding areas suggests they could maintain coherence for time-sensitive processes. Meanwhile, the entropy variability in non-coding regions implies they might modulate coherence dynamically based on external signals. Together, this dual system reflects how DNA might measure time dynamically, adapting to different signals.

The observed quantum connection between regions (measured as quantum entanglement) supports the idea that DNA functions as an adaptive quantum system, where both region types interact to maintain coherence. Non-coding regions, with their higher variability, could act like sensors receiving cosmic wave shifts. These perturbations are entangled with the coding regions, enabling the DNA to adapt its biological rhythms to the universe's temporal flow. The system's design resembles quantum computing principles, where stability (coding regions) and adaptability (non-coding regions) work together.

Recent advances in photonic quantum frequency combs have demonstrated how complex internal modal structures can support high-dimensional entanglement, enabling scalable qudit-based quantum computation and communication systems [37]. It could be possible that DNA's hierarchical and fractal molecular architecture may offer an analogous platform, with non-coding and coding regions forming dynamically entangled subspaces across multiple biological scales. High-dimensional quantum key distribution (QKD) uses large-alphabet encoding to improve secure information transfer by maximizing entropy per symbol [38]. Similarly, the results show that DNA exhibits distinct entropy patterns between coding and non-coding regions, which may function as entropic domains for information flow. The observed entanglement and entropy gradients suggest that DNA could behave as a natural, passive quantum communication channel, redistributing information in response to external signals, similar to how QKD systems reconcile entropy across sender and receiver.

When simulating DNA implicit perturbations over time steps, the amplitude differences reinforce the idea that DNA may use coding and non-coding regions differently in quantum terms. The phase coherence suggests that DNA, as a quantum system, maintains internal synchronization across regions, enabling it to function as a cohesive computational unit rather than as isolated segments. Moreover, when simulating DNA external perturbations, the coding regions are "in sync" with the oscillatory behavior of the non-coding regions, potentially allowing, in the model, the information encoded in the frequency and phase of the incoming cosmic signal to be "transferred" through this coherence.

The simulations of external shifted cosmic perturbations over DNA as a quantum system regarding the quantum tunneling probabilities provide intriguing evidence supporting the idea that DNA's coding and non-coding regions have distinct quantum mechanical properties, linked to cosmic interactions. The transition in the simulations from an exaggerated Doppler shift (proof of concept) to a realistic value ($\Delta f = 0.0008$ Hz) demonstrates the feasibility of DNA's response to weak cosmic signals under biologically relevant conditions. Even though the perturbation amplitudes in the simulation are extremely small and may seem negligible, the consistent statistical differences observed in the real sequences suggest that the DNA's intrinsic properties lead to differential quantum behaviors between coding and non-coding regions. This implies that the model, despite its limitations, captures some aspect of the DNA's quantum mechanical behavior that aligns with the main hypothesis. Stochastic resonance effectively amplifies weak cosmic signals, demonstrating that biologically realistic noise can overcome decoherence, supporting the hypothesis without assuming idealized quantum

coherence in DNA's quantum dynamics. Therefore, any differences observed are likely due to the sequences themselves rather than the noise. The statistical significance in real sequences, but not in controls, indicates that noise did not disproportionately affect one region over the other. The model serves as a proof of concept, demonstrating that differential quantum behaviors can be observed between coding and non-coding regions under certain conditions. It provides a foundation for further, more detailed studies. Observing differences in tunneling probabilities suggests potential implications for mutation rates and biological processes like time measures in cellular aging and genome evolution. In contemporary mutation frameworks, replication errors are treated as rare stochastic events whose effective rate reflects polymerase kinetics, local sequence context, proofreading, and post-replicative mismatch repair. In this view, point mutations arise when a transient mispairing event occurs during nucleotide incorporation and subsequently escapes correction. The mechanistic link to quantum tunnelling is that proton transfer along Watson–Crick hydrogen bonds (and related intrabase transfers) can generate short-lived tautomeric configurations that alter base-pairing preferences. Certain tautomers can maintain near Watson–Crick-like geometry while encoding an incorrect hydrogen-bonding pattern, which can increase the probability of misincorporation without requiring large helical distortions that would necessarily trigger immediate rejection [4,39]. The key "microtiming" aspect is that polymerase nucleotide selection and active-site closure define a finite kinetic window during which the instantaneous hydrogen-bond donor/acceptor pattern of the templating base biases dNTP selection. If a tautomeric configuration is occupied within this window, misincorporation becomes more likely. Subsequent relaxation of the base to its canonical form converts a transient quantum event into a classical mismatch. Fixation then depends on downstream processes) proofreading and mismatch repair (whose efficacy is time- and context-dependent; hence the overall mutation probability can be expressed as the product of an instantaneous misincorporation hazard and a repair-escape probability, both already formalized within stochastic and context-dependent mutation models. Within this established probabilistic landscape, the results in this work can be interpreted as evidence for a physically plausible modulation mechanism: in the tunnelling module, a weak time-dependent perturbation combined with physiological-like noise modulates the instantaneous tunnelling probability in an effective double-well model. Importantly, the modulation is sequence-dependent (real genomic segments differ from shuffled controls) and coupled across regions through the joint Hamiltonian, providing a route by which perturbation-sensitive dynamics in non-coding segments could influence coupled dynamics in coding segments. This does not imply deterministic "control" of mutations; rather, it supports the narrower claim that external structured perturbations can act as weak modulators of the tautomerisation/tunnelling hazard function, thereby biasing the long-run distribution of replication errors across many incorporation events, consistent with stochastic mutation theory.

While the external drive is parametrised using Doppler-shift-inspired frequencies, the present simulations do not constitute experimental evidence of cosmological influence on DNA. Instead, they demonstrate that, under noisy, physiological-like conditions, real genomic sequences can show a non-trivial, Doppler-sensitive response that reduced or not reproduced when the ordering/correlation structure of the input is disrupted by control constructions, which is the hallmark of matched-filter-like behaviour in this toy model.

Coding and non-coding DNA regions are known to differ in classical statistical properties, including nucleotide composition, GC bias, and codon-associated periodicity. These differences define the initial conditions of the modeled system and are therefore expected to influence any subsequent dynamical evolution, classical or quantum. Accordingly, a separation between coding and non-coding trajectories (or between real and shuffled sequences) is not surprising per se; the purpose of the dynamical analysis is to determine which sequence-organizational features are required for the observed time-dependent responses under the specified Hamiltonian. The aim of the present work is not to interpret such static sequence statistics as evidence of quantum processing per se. Instead, the focus is on dynamical observables (including time-dependent phase evolution, spectral power redistribution under structured perturbations, inter-region amplitude transfer, and modulation of proton tunnelling probabilities) which arise only after propagation under a Hamiltonian with explicit temporal structure. Shuffled controls are used to destroy long-range and context-dependent correlations while

preserving gross composition, thereby testing whether the observed dynamical responses depend on biologically meaningful sequence organization rather than on random fluctuations. Importantly, the presence of a difference between real and shuffled sequences is not, by itself, interpreted as proof of quantum behavior; rather, it establishes that sequence structure conditions the system's response to time-dependent perturbations, which is a prerequisite for the quantum-sensitivity mechanisms explored here.

In quantum computation, gates are unitary operations that transform quantum states in a well-defined and reversible manner. Within the present model, DNA does not rely on externally applied gate pulses, as in conventional quantum circuits. Instead, the natural Hamiltonian dynamics of the DNA molecule itself — including base-specific energies and nearest-neighbor couplings — define a time evolution operator ($U = \exp(-iHt/\hbar)$) that effectively behaves as a sequence of quantum gates. For instance, differences in base binding energies act as phase shift gates, while the tridiagonal coupling structure can produce entanglement, akin to controlled-NOT or iSWAP gates. Additionally, the application of external time-dependent fields (e.g., 34 GHz Doppler-shifted signals) modifies the Hamiltonian in a dynamic way, introducing gate-like transformations across specific regions. In this sense, DNA encodes and applies quantum gates *endogenously*, with the entire molecule acting as a naturally evolving quantum circuit.

Overall, these results support the conceptual notion that DNA's quantum structure measure cosmic time by capturing Doppler-shifted signals, reframing the quantum system based on entropic input, and translating time-measure information from cosmic to biological scale, altering mutations probability occurrence via quantum tunneling. While further experimental evidence is needed (see Supporting information, Appendix A: study limitations), this work proposes a novel conceptual frame to understand nature. A concise equation summarizing the interplay between quantum entropy, cosmic frequency shifts, and DNA quantum dynamics is provided in the supplementary information.

Professor Lenski and collaborators have maintained a continuous culture of Escherichia coli for over 30 years, spanning more than 75,000 generations [5]. These bacteria replicate their DNA multiple times per day, making this one of the most comprehensive experimental models for studying evolution. The experiment explores how populations adapt to various selection pressures, such as nutrient availability and temperature. Remarkably, even in lineages not directly exposed to such pressures, adaptive mutations and functional innovations have emerged. Some of these evolutionary leaps appear difficult to explain by classical models alone, highlighting the need for new theoretical frameworks to understand such complexity [40]. Current evolutionary models face limitations in explaining how mutations emerge and propagate beyond purely stochastic mechanisms [6]. While mutations are central to evolution, their occurrence, distribution, and potential biases remain poorly understood. Additionally, the role of external environmental factors, particularly electromagnetic and quantum-scale influences, in mutation dynamics is largely unexplored. Addressing these gaps is crucial for advancing our understanding of evolutionary processes and genetic adaptation.

Previous works have related the isolation of organisms from electromagnetic fields with the alteration of basic biological functions [41,42]. Diaz del Cerro et. al showed that the use of a bed with an insulating system of electromagnetic fields improves immune function, redox, and inflammatory states, and decreases the rate of aging [43]. The results obtained in this study are consistent with experimental evidence showing the impact of electromagnetic radiation on genomic stability. Studies have demonstrated increased DNA damage and mutational activity under specific electromagnetic conditions, particularly in bacterial and eukaryotic systems [44–46].

The proposed model in this work is a concept supported by quantum walk frameworks, which reveal that quantum-driven genotype exploration is potentially more efficient than classical random walks, especially at scales where coherence dominates [27]. Moreover, the continuous-time quantum walk model with Poisson-distributed measurements [28] provides insights into how DNA might probabilistically sample environmental inputs such as cosmic radiation, reflecting realistic interactions. Additionally, quantum statistical approaches that incorporate environmental stressors, like radiation and temperature, to model DNA mutations [10], directly align with the hypothesis that DNA mutations could represent quantum-based measurements of universal expansion. Finally, Kurian's considerations on life's computational limits

 

within universal boundaries [47] further validate the proposal that DNA might indeed process universal-scale informational entropy, positioning biological systems within broader cosmological computational constraints.

This hypothesis, while speculative, provides a conceptual foundation supported by computational models and mathematical analysis, encouraging experimental studies to further explore these possibilities and validate the interplay between the quantum properties of DNA and universal phenomena. One interesting approach to experimentally corroborate these ideas would be to study the relevance of DNA as a quantum system in bacteria evolution experiments. In a recent study, Bena et. al. developed a real-time imaging system to track the fate of spontaneous replication errors in Escherichia coli [1]. The research revealed that many spontaneous mutations arise from errors detected by mismatch repair but are inefficiently repaired due to a temporal constraint imposed by DNA strand methylation. A similar approach could be used to analyze the role of DNA as a quantum computer in the process of mutation accumulation. If the hypothesis is correct, growing the bacteria in and out of a Faraday cage, would be enough to change the mutation dynamics observed by the described method in Bena et. al.'s work. With the knowledge provided in the present work, in silico models and simulations can be created to "predict" the behavior of DNA as a quantum computer. Thus, by modifying the bacterial genome in non-coding sequences with CRISPR technology, and in parallel simulating the mutation evolution dynamics of WT vs. CRISPR genomes (in and out of Faraday cages), it would be possible to in silico predict the DNA mutation evolution, and corroborate these predictions in the wet lab, validating the main hypothesis of the present work.

## Supporting information

**S1 File. Supporting extended materials and methods section.**
(DOCX)

**S1 Table. Quantum evolution over time with Doppler effect for real 200 bp sequence.**
(DOCX)

**S2 Table. Quantum evolution over time with Doppler effect for control 200 bp sequence.**
(DOCX)

**S3 Table. Simulations for mutation induction by quantum tunneling.**
(DOCX)

**S4 Table. Simulations for mutation induction by quantum tunneling in human CRY1 gene.**
(DOCX)

## Author contributions

**Conceptualization:** Nahuel Aquiles Garcia.

**Data curation:** Nahuel Aquiles Garcia.

**Formal analysis:** Nahuel Aquiles Garcia.

**Investigation:** Nahuel Aquiles Garcia.

**Methodology:** Nahuel Aquiles Garcia.

**Software:** Nahuel Aquiles Garcia.

**Writing – original draft:** Nahuel Aquiles Garcia.

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
