## [Decision Letter · Decision Letter 0]

31 Jan 2026

Dear Dr. Garcia,

Thank you for submitting your manuscript to PLOS ONE. After careful consideration, we feel that it has merit but does not fully meet PLOS ONE’s publication criteria as it currently stands. Therefore, we invite you to submit a revised version of the manuscript that addresses the points raised during the review process.

We look forward to receiving your revised manuscript.

Kind regards,

Yang (Jack) Lu, PhD

Academic Editor

PLOS One

Journal Requirements:

3. Thank you for stating the following in the Competing Interests/Financial Disclosure section:

The authors have declared that no competing interests exist.

We note that one or more of the authors are employed by a commercial company: GECORP

Reviewers' comments:

Reviewer's Responses to Questions

**Comments to the Author**

1. Is the manuscript technically sound, and do the data support the conclusions?

Reviewer #1: No

Reviewer #2: Yes

2. Has the statistical analysis been performed appropriately and rigorously?

Reviewer #1: N/A

Reviewer #2: Yes

3. Have the authors made all data underlying the findings in their manuscript fully available?

Reviewer #1: Yes

Reviewer #2: Yes

4. Is the manuscript presented in an intelligible fashion and written in standard English?

Reviewer #1: Yes

Reviewer #2: Yes

Reviewer #1: The discussion presented in the paper is both interesting and valuable; however, several sections require clarification and improvement, both technically and substantively.

Throughout the manuscript, there are numerous typographical inconsistencies, particularly in the formatting of enumerated lists and the notation of mathematical symbols. For example, the symbol “\lambda i” (page 11) should be consistently written as “\lambda_i” to ensure mathematical correctness and readability. I recommend a thorough review of the manuscript to unify the typographical style and correct such issues.

The main issue with the paper is the lack of mathematical rigor in the presentation of results. In several places, transformations are described using Python code fragments. While code can be helpful in illustrating implementation, the main body of the paper should provide formal mathematical descriptions of the formulas and transformations discussed. The code itself may be included in a later section dedicated to the simulation framework, but the theoretical part requires precise mathematical definitions and explanations.

The mapping between DNA sequences and qubits, as presented (section 3.2), is currently limited to only four basic cases (the collapsed state for guanine is also not explained, how such state is constructed). Given that the space of qubits is, in principle, infinite, the current approach does not convincingly demonstrate the general applicability of the proposed simulation model. I recommend expanding this section to explain how the mapping can be generalized to arbitrary DNA sequences and to provide a more comprehensive justification for the chosen approach.

Another area that requires improvement is the calculation of von Neumann (quantum) entropy. The paper does not clearly explain how such entropies are computed using DNA encoding. For instance, the statement “Each nucleotide in the DNA sequence is mapped to a complex amplitude, representing its quantum state” is not sufficiently detailed. It is unclear how the mapping from DNA to complex amplitudes is performed. I suggest providing a detailed algorithm or mathematical description of this process, accompanied by illustrative examples.

In its current form, the paper requires substantial effort to clarify the text and to improve the mathematical aspects of the description of DNA encoding for quantum computation models. Introducing formal algorithms and precise mathematical explanations will facilitate future verification of the results. Nevertheless, I appreciate the valuable ideas presented in the work. The included appendix provides additional information relevant to the main text. Although the paper does not present a full mathematical analysis, the concepts introduced are interesting and merit broader dissemination. I also recommend addressing remaining technical issues, such as replacing equations presented as images with proper mathematical text, to further enhance the clarity and quality of the manuscript.

Reviewer #2: Strengths：

1. Novelty and Originality: The attempt to link cosmological phenomena (universe expansion/Doppler shifts) with molecular biological aging via quantum tunneling is a highly unique and creative hypothesis.

2. Computational Transparency: The author is transparent about the "toy model" nature of the work and acknowledges significant physical limitations, such as the attenuation of 34 GHz signals by the atmosphere and tissue.

3. Statistical Rigor in Simulations: The manuscript employs a robust set of statistical comparisons (T-tests, Mann-Whitney U) between real genomic sequences and randomized controls to validate the internal consistency of the simulation results.

4. Clear Organization: The distinction between the Shannon entropy analysis, entanglement analysis, and tunneling simulations is logically structured.

Issues:

1. Conflation of Sequence Statistics with Quantum Processing: The paper claims that real genomic segments exhibit distinctive dynamical signatures compared to controls. However, coding regions have specific statistical properties (codon periodicity, GC bias) that non-coding regions lack. The simulation likely amplifies these classical statistical differences. The author needs to demonstrate that the "quantum" differences are not simply artifacts of feeding different classical statistical distributions into a differential equation. The "shuffled controls" remove these statistical correlations, so a difference is expected regardless of quantum effects.

2. Fractal Antenna Terminology: The use of "DNA as a fractal antenna" is treated as a phenomenological shorthand. However, the manuscript cites controversial papers (e.g., Blank & Goodman) without critically addressing the lack of replication in the broader biophysics community regarding DNA microwave resonance.

**Do you want your identity to be public for this peer review?** For information about this choice, including consent withdrawal, please see our Privacy Policy

Reviewer #1: No

Reviewer #2: **Yes:** Ziyu Wang

---

## [Author Response · Author response to Decision Letter 1]

9 Feb 2026

Revised submission for PONE-D-26-00422 | “DNA as a Quantum System in Evolution”

Response to reviewers

Reviewer 1: The discussion presented in the paper is both interesting and valuable; however, several sections require clarification and improvement, both technically and substantively.

Thank you sincerely for your careful and constructive assessment. I appreciate your recognition of the value and potential impact of the ideas presented. Following your recommendations, I have substantially clarified the text and strengthened the mathematical description of the DNA encoding by adding formal algorithms and precise definitions. Please feel free to suggest any additional changes you would like to see, your feedback is very welcome.

Throughout the manuscript, there are numerous typographical inconsistencies, particularly in the formatting of enumerated lists and the notation of mathematical symbols. For example, the symbol “\lambda i” (page 11) should be consistently written as “\lambda_i” to ensure mathematical correctness and readability. I recommend a thorough review of the manuscript to unify the typographical style and correct such issues.

I thank the reviewer for the careful reading and for highlighting typographical inconsistencies. I performed a comprehensive typographical audit of the entire manuscript and Supplementary Materials, focusing on (i) consistent formatting of enumerated lists and procedural steps, and (ii) consistent mathematical notation. In particular, I standardized all indexed symbols to a single, consistent subscript notation throughout (e.g., eigenvalues λ_i in the von Neumann/entanglement entropy definitions and nucleotide probabilities p(x_i) in Shannon entropy), and I harmonized quantum-state notation (e.g., |ψ_i⟩, |ϕ_j⟩) and density-matrix/trace notation (e.g., ρ_A, Tr_B(ρ_joint)). I also corrected minor spacing/punctuation artifacts (e.g., stray double punctuation) and unified list formatting across sections. These changes are typographical only and do not alter any results, analyses, or conclusions.

The main issue with the paper is the lack of mathematical rigor in the presentation of results. In several places, transformations are described using Python code fragments. While code can be helpful in illustrating implementation, the main body of the paper should provide formal mathematical descriptions of the formulas and transformations discussed. The code itself may be included in a later section dedicated to the simulation framework, but the theoretical part requires precise mathematical definitions and explanations.

I thank the reviewer for this important point and fully agree. In the revised manuscript, I replaced implementation-style Python code fragments in the main text with formal mathematical definitions of the objects and transformations used in the simulations. Specifically, I now provide an explicit mathematical formulation of: (i) the state representation in a site-basis Hilbert space, (ii) the initialization map from nucleotides to complex amplitudes, (iii) the block-structured total Hamiltonian (coding ⊕ non-coding), (iv) the boundary coupling term connecting coding and non-coding regions, (v) the time-dependent perturbation applied to the non-coding block (including the Doppler-shifted driving frequency), and (vi) the time-dependent Schrödinger equation and the discrete numerical update rule used in the simulations (including normalization to control numerical drift).

To preserve reproducibility without reducing mathematical clarity in the main manuscript, the corresponding Python implementation snippet(s) were moved to the Supplementary Methods under a dedicated “Implementation: Code Workflow” subsection. This keeps the theoretical development formal and self-contained while still providing concrete computational details for readers who want to reproduce the simulations. Moreover, please notice that because of a size/journal formatting issue, a full mathematical description of the applied models is provided between both sections (main manuscript and supplementary information).

The mapping between DNA sequences and qubits, as presented (section 3.2), is currently limited to only four basic cases (the collapsed state for guanine is also not explained, how such state is constructed). Given that the space of qubits is, in principle, infinite, the current approach does not convincingly demonstrate the general applicability of the proposed simulation model. I recommend expanding this section to explain how the mapping can be generalized to arbitrary DNA sequences and to provide a more comprehensive justification for the chosen approach.

Thank you for this super important point. I agree that my earlier wording could be interpreted as limiting generality, so I have expanded Section 3.2 to make the scope and construction explicit.

Generality to arbitrary DNA sequences. I clarified that the encoding is a deterministic feature map from the DNA alphabet {A, C, G, T} into a low-dimensional quantum state space and is applied site-by-site. Therefore, it is defined for arbitrary DNA sequences of any length and composition; the “four cases” simply correspond to the four nucleotide symbols, not a restriction on sequence generality.

Clarification of the guanine “collapsed” state. I clarified that the vector [0,0] is not a normalized qubit; in this model it is used as a shorthand for an erasure/vacuum state (denoted |vac>) orthogonal to {|0>,|1>}. Computationally, guanine positions are initialized with zero amplitude and the full sequence state (or density matrix) is normalized, so guanine contributes a zero projector at initialization. Conceptually, this represents a strongly damped/background degree of freedom under the chosen coarse-graining (i.e., a rapidly relaxed/dephased contribution that does not carry coherent amplitude in the selected two-level subspace).

Explicit generalization to the full (continuous) qubit space. To address the reviewer’s point that qubit state space is continuous, I added an explicit parameterized mapping option (described in the Supplementary Methods and referenced in the main text): for each base b∈{A,C,G,T}, one may define

|ψ(b)⟩ = cos(θ_b/2) |0⟩ + e^(i φ_b ) sin(θ_b/2) |1⟩

where (thetab, phib) can be fixed constants or derived from normalized biochemical

descriptors (e.g., stacking energies, hydrogen-bond pattern, dipole moment, oxidation potential), and optionally extended to local k-mer context. This makes clear how the mapping can be generalized beyond a discrete subset of states if desired.

Justification for the discrete encoding used in the main simulations. I also added a short justification: although a qubit admits a continuum of pure states, DNA in vivo is an open quantum system, so environmental coupling is expected to decohere most superpositions rapidly, effectively selecting a restricted set of robust pointer-like states. I therefore start from a minimal, interpretable encoding (avoiding unnecessary free parameters) while allowing Hamiltonian time evolution to generate continuous complex amplitudes and phases.

Another area that requires improvement is the calculation of von Neumann (quantum) entropy. The paper does not clearly explain how such entropies are computed using DNA encoding. For instance, the statement “Each nucleotide in the DNA sequence is mapped to a complex amplitude, representing its quantum state” is not sufficiently detailed. It is unclear how the mapping from DNA to complex amplitudes is performed. I suggest providing a detailed algorithm or mathematical description of this process, accompanied by illustrative examples.

Thank you for this important suggestion. I have expanded the manuscript and

Supplementary Methods to explicitly define the DNA-to-quantum-state encoding used for von Neumann entropy and to clarify how the DNA sequence is mapped into complex amplitudes for the time-dependent Schrödinger simulations. Specifically, I now provide: (i) an explicit site-wise mapping from nucleotides {A,C,G,T} to normalized qubit vectors (A→|0⟩, T→|1⟩, C→(|0⟩+|1⟩)/√2, and G treated as an erasure/vacuum state), (ii) a step-by-step density-matrix construction as the average of per-site projectors followed by trace normalization, (iii) the von Neumann entropy evaluation from the eigenvalues of the density matrix, and (iv) an illustrative worked example (“ATC”) and an algorithmic pseudocode description. In addition, I clarify the equivalent scalar complex-amplitude initialization used in the Hamiltonian evolution (α(A)=1, α(T)=i, α(C)=(1+i)/√2, α(G)=0, followed by normalization), and explain its correspondence to the qubit encoding. These additions provide a complete and reproducible description of the entropy computation and the DNA encoding used throughout the work.

Reviewer 2

Conflation of Sequence Statistics with Quantum Processing: The paper claims that real genomic segments exhibit distinctive dynamical signatures compared to controls. However, coding regions have specific statistical properties (codon periodicity, GC bias) that non-coding regions lack. The simulation likely amplifies these classical statistical differences. The author needs to demonstrate that the "quantum" differences are not simply artifacts of feeding different classical statistical distributions into a differential equation. The "shuffled controls" remove these statistical correlations, so a difference is expected regardless of quantum effects.

I thank the reviewer for raising this important point and I fully agree that coding and non-coding regions differ in well-known classical statistical properties (e.g., GC bias, codon-related periodicity, and higher-order correlations), and that shuffled controls disrupt these correlations, so a difference between real and shuffled sequences can be expected in any order-sensitive dynamical system, independent of any “quantum” interpretation. In the current way, it is confusing, thanks for point it out. I have therefore clarified in the revised manuscript that I do not interpret static sequence statistics (or real-vs-shuffled differences alone) as evidence of quantum processing. Rather, sequence statistics define initial conditions, while the analysis focuses on dynamical observables that arise only after propagation under an explicitly time-dependent Hamiltonian (time-dependent phase evolution, spectral power redistribution under structured perturbations, inter-region amplitude transfer via coupling, and modulation of proton tunnelling probabilities). I also explicitly state the role and limitations of simple shuffled controls: they test sensitivity to sequence organization but do not, by themselves, constitute proof of quantum effects; more stringent surrogate controls (e.g., di-nucleotide–preserving and codon-phase–preserving surrogates) and classical dynamical baselines are appropriate next steps to further isolate which sequence features drive the observed dynamical separation.

In the new version, the reported “distinctive dynamical signatures” should be interpreted as sequence-dependent signatures in a quantum-like evolution model, not as proof that biology is “performing quantum computation” simply because coding and non-coding differ. Importantly, I now make explicit that the purpose of the “control/shuffle” procedures is to test whether the observed response depends only on low-order composition (expected to differ between region types) or instead on ordering/correlation structure beyond composition. Accordingly, I (i) define the control construction precisely in the Supplementary Methods, (ii) remove/replace ambiguous wording (“shuffled controls”) where it could be misread as evidence of uniquely quantum effects, and (iii) add an explicit limitation noting that composition- and correlation-matched surrogate controls (e.g., mono-/di-nucleotide preserving shuffles; codon-phase–preserving surrogates for coding segments) are the appropriate next step to further isolate which specific statistical features drive the dynamical separation.

Fractal Antenna Terminology: The use of "DNA as a fractal antenna" is treated as a phenomenological shorthand. However, the manuscript cites controversial papers (e.g., Blank & Goodman) without critically addressing the lack of replication in the broader biophysics community regarding DNA microwave resonance.

Thanks for this constructive comment. Please note that I cite the “DNA as a fractal antenna” in two points (I agree that there is not enough criticism, as you suggested):

Introduction (Main hypothesis ): paragraph titled “Experimental basis for DNA–EM coupling and the ‘antenna’ metaphor.” In this paragraph I state that evidence for DNA–EM coupling is “mixed but non-empty,” I cite both supportive and unfavorable reports, and I explicitly frame the term as phenomenology: “Accordingly, ‘DNA as a fractal antenna’ is best treated as a phenomenological shorthand…” (followed by the clarification that effective coupling is dominated by dielectric microenvironment and hydration conditions rather than bases alone).

Appendix A (“Study Limitations” | opening paragraph). Here I again treat the “antenna” language as motivation rather than a literal in-vivo resonance mechanism, explicitly noting that 34–160 GHz photons are strongly attenuated by atmosphere and tissue and that the local field near DNA is dominated by thermal/anthropogenic RF backgrounds; I therefore treat the external drive as an abstract weak perturbation with a slow Doppler-like temporal structure, not as a quantitatively calibrated claim of direct CMB coupling.

Regarding the specific mention of Blank & Goodman, I cite those papers for completeness and transparency alongside additional evidence (including length-dependent resonant absorption reports and THz spectroscopy signatures), and I also include null/negative dielectric findings to avoid cherry-picking. The manuscript’s central modeling results and conclusions do not depend on any single contested study, nor do they require a literal 34 GHz in-vivo resonance; the 34 GHz choice is used as a representative GHz-scale motif to instantiate a weak structured drive and generate testable predictions. To add extra clarification over this I have included a new paragraph: “I cite these studies for context and transparency; however, reported GHz responses are condition-dependent and not uniformly reproduced across setups, so here ‘fractal antenna’ denotes multiscale EM coupling rather than a definitive in-vivo resonance claim”.

Thanks again for these constructive comments and for helping me improve the clarity and rigor of the manuscript. Please feel free to suggest any further modifications or additional checks you think would strengthen the work, I will be happy to implement them.

---

## [Decision Letter · Decision Letter 1]

23 Feb 2026

DNA as a Quantum System in Evolution

PONE-D-26-00422R1

Dear Dr. Garcia,

We’re pleased to inform you that your manuscript has been judged scientifically suitable for publication and will be formally accepted for publication once it meets all outstanding technical requirements.

Kind regards,

Yang (Jack) Lu, PhD

Academic Editor

PLOS One

Additional Editor Comments (optional):

Reviewers' comments:

Reviewer's Responses to Questions

**Comments to the Author**

Reviewer #1: All comments have been addressed

Reviewer #2: All comments have been addressed

2. Is the manuscript technically sound, and do the data support the conclusions?

Reviewer #1: Yes

Reviewer #2: Partly

3. Has the statistical analysis been performed appropriately and rigorously?

Reviewer #1: Yes

Reviewer #2: Yes

4. Have the authors made all data underlying the findings in their manuscript fully available?

Reviewer #1: Yes

Reviewer #2: Yes

5. Is the manuscript presented in an intelligible fashion and written in standard English?

Reviewer #1: Yes

Reviewer #2: Yes

Reviewer #1: Changes made by the Author to the current version should be assessed as correct and improve the overall evaluation of the work.

Reviewer #2: (No Response)

**Do you want your identity to be public for this peer review?** For information about this choice, including consent withdrawal, please see our Privacy Policy

Reviewer #1: No

Reviewer #2: No

---

## [Editor Report · Acceptance letter]

PONE-D-26-00422R1

PLOS One

Dear Dr. Garcia,

I'm pleased to inform you that your manuscript has been deemed suitable for publication in PLOS One. Congratulations! Your manuscript is now being handed over to our production team.

Kind regards,

on behalf of

Dr. Yang (Jack) Lu

Academic Editor

PLOS One